# LATENT ABSTRACTIONS IN GENERATIVE DIFFUSION MODELS

## ABSTRACT

In this work we study how diffusion-based generative models produce high-dimensional data, such as an image, by implicitly relying on a manifestation of a low-dimensional set of latent abstractions, that guide the generative process. We present a novel theoretical framework that extends Nonlinear Filtering (NLF), and that offers a unique perspective on SDE-based generative models. The development of our theory relies on a novel formulation of the joint (state and measurement) dynamics, and an information-theoretic measure of the influence of the system state on the measurement process. According to our theory, diffusion models can be cast as a system of SDE, describing a non-linear filter in which the evolution of unobservable latent abstractions steers the dynamics of an observable measurement process (corresponding to the generative pathways). In addition, we present an empirical study to validate our theory and previous empirical results on the emergence of latent abstractions at different stages of the generative process.

## 1 INTRODUCTION

Generative models have become a cornerstone of modern machine learning, offering powerful methods for synthesizing high-quality data across various domains such as image and video synthesis (Dhariwal & Nichol, 2021; Ho et al., 2022; He et al., 2022), natural language processing (Li et al., 2022b; He et al., 2023; Gulrajani & Hashimoto, 2023; Lou et al., 2024), audio generation (Kong et al., 2021; Liu et al., 2022), and molecular structures and general 3D shapes (Trippe et al., 2022; Hoogeboom et al., 2022; Luo & Hu, 2021; Zeng et al., 2022), to name a few. These models transform an initial distribution, which is simple to sample from, into one that approximates the data distribution. Among these, diffusion-based models designed through the lenses of Stochastic Differential Equations (SDEs) (Song et al., 2021; Ho et al., 2020; Albergo et al., 2023) have gained popularity due to their ability to generate realistic and diverse data samples through a series of stochastic transformations.

In such models, the data generation process, as described by a substantial body of empirical research (Chen et al., 2023; Linhardt et al., 2024; Tang et al., 2023), appears to develop according to distinct stages: high-level semantics emerge first, followed by the incorporation of low-level details, culminating in a refinement (denoising) phase. Despite ample evidence, a comprehensive theoretical framework for modeling these dynamics remains underexplored. Indeed, despite recent work on SDE-based generative models (Berner et al., 2022; Richter & Berner, 2023; Ye et al., 2022; Raginsky, 2024) shed new lights on such models, they fall short of explicitly investigating the emergence of abstract representations in the generative process. We address this gap by establishing a new framework for elucidating how generative models construct and leverage latent abstractions, approached through the paradigm of NLF (Bain & Crisan, 2009; Van Handel, 2007; Kutschireiter et al., 2020).

NLF is used across diverse engineering domains (Bain & Crisan, 2009), as it provides robust methodologies for the estimation and prediction of a system's state amidst uncertainty and noise. NLF enables the inference of dynamic latent variables that define the system state based on observed data, offering a Bayesian interpretation of state evolution and the ability to incorporate stochastic system dynamics. The problem we consider is the following: an *unobservable* random variable $X$ is measured through a noisy continuous-time process $Y_t$, wherein the influence of $X$ on the noisy process is described by an observation function $H$, with the noise component modeled as a Brownian motion term. The goal is to estimate the a-posteriori measure $\pi_t$ of the variable $X$ given the entire historical trajectory of the measurement process $Y_t$.

In this work, we establish a connection between SDE-based generative models and NLF by observing that they can be interpreted as *simulations* of NLF dynamics. In our framework, the latent abstraction, which corresponds to certain real-world properties within the scope of classical nonlinear filtering and remains unaffected in a *causal* manner by the posterior process $\pi_t$, is implicitly simulated and iteratively refined. We explore the connection between latent abstractions and the a-posteriori process, through the concept of *filtrations* – broadly defined as collections of progressively increasing information sets – and offer a rigorous theory to study the emergence and influence of latent abstractions throughout the data generation process. Our theoretical contributions unfold as follows.

In § 2 we show how to reformulate classical NLF results such that the measurement process is the only available information, and derive the corresponding dynamics of both the latent abstraction and the measurement process. These results are summarized in Theorem 2 and Theorem 3.

Given the new dynamics, in Theorem 4 we show how to estimate the a-posteriori measure of the NLF model, and present a novel derivation to compute the mutual information between the measurement process and random variables derived from a transformation of the latent abstractions in Theorem 5. Finally, we show in Theorem 6, that the a-posteriori measure is a sufficient statistics for any random variable derived from the latent abstractions, when only having access to the measurement process.

Building on these general results, in § 3 we present a novel perspective on continuous-time score-based diffusion models, which is summarised in Equation (10). We propose to view such generative models as NLF simulators that progress in two stages: first our model updates the a-posteriori measure representing a sufficient statistics of the latent abstractions, second, it uses a projection of the a-posteriori measure to update the measurement process. Such intuitive understanding is the result of several fundamental steps. In Theorem 7 and Theorem 8, we show that the common view of score-based diffusion models by which they evolve according to forward (noising) and backward (generative) dynamics is compatible with the NLF formulation, in which there is no need to distinguish between such phases. In other words, the NLF perspective of Equation (10) is a valid generative model. In Appendix H, we provide additional results (see Lemma 1), focusing on the specific case of linear diffusion models, which are the most popular instance of score-based generative models in use today. In § 4, we summarize the main intuitions behind our NLF framework.

Our results explain, by means of a theoretically sound framework, the emergence of latent abstractions that has been observed by a large body of empirical work (Bisk et al., 2020; Bender & Koller, 2020; Li et al., 2022a; Park et al., 2023; Kwon et al., 2023; Chen et al., 2023; Linhardt et al., 2024; Tang et al., 2023; Xiang et al., 2023; Haas et al., 2024). The closest research to our findings is discussed in (Sclocchi et al., 2024), albeit from a different mathematical perspective. To root our theoretical results in additional empirical evidence, we conclude our work in § 5 with a series of experiments on score-based generative models (Song et al., 2021), where we 1) validate existing probing techniques to measure the emergence of latent abstractions, 2) compute the mutual information as derived in our framework, and show that it is a suitable approach to measure the relation between the generative process and latent abstractions, 3) introduce a new measurement protocol to further confirm the connections between our theory, and how practical diffusion-based generative models operate.

## 2  NONLINEAR FILTERING

Consider two random variables $Y_t$ and $X$, corresponding to a stochastic **measurement** process ($Y_t$) of some underlying **latent abstraction** ($X$). We construct our universe sample space $\Omega$ as the combination of the space of continuous functions in the interval $[0, T]$ ($T \in \mathbb{R}^+$) and of a complete separable metric space $\mathcal{S}$, i.e., $\Omega = \mathcal{C}([0,T], \mathbb{R}^N) \times \mathcal{S}$. On this space, we consider the joint *canonical* process $Z_t(\omega) = [Y_t, X] = [\omega_t^y, \omega^x]$ for all $\omega \in \Omega$, with $\omega = [\omega^y, \omega^x]$. In this work we indicate with $\sigma(\cdot)$ sigma-algebras. Consider the growing filtration naturally induced by the canonical process $\mathcal{F}_t^{Y,X} = \sigma(Y_{0 \leq s \leq t}, X)$ (a short-hand for $\sigma(\sigma(Y_{0 \leq s \leq t}) \cup \sigma(X))$), and define $\mathcal{F} = \mathcal{F}_T^{Y,X}$. We build the probability triplet $(\Omega, \mathcal{F}, \mathrm{P})$, where the probability measure $\mathrm{P}$ is selected such that the process $\{Z_{0 \leq t \leq T}, \mathcal{F}_{0 \leq t \leq T}^{Y,X}\}$ has the following SDE representation

$$Y_t = Y_0 + \int_0^t H(Y_s, X, s)\mathrm{d}s + W_t, \tag{1}$$

where $\{W_{0 \le t \le T}, \mathcal{F}_{0 \le t \le T}^{Y,X}\}$ is a Brownian motion with initial value 0 and $H : \Omega \times [0, T] \to \mathbb{R}^N$ is an *observation* process. All standard technical assumptions are available in Appendix A.

Next, we provide the necessary background on NLF, to pave the way for understanding its connection with the generative models of interest. The most important building block of the NLF literature is represented by the **conditional probability measure** $\mathrm{P}[X \in A \mid \mathcal{F}_t^Y]$ (notice the reduced filtration $\mathcal{F}_t^Y \subset \mathcal{F}_t^{Y,X}$), which summarizes, a-posteriori, the distribution of $X$ given observations of the measurement process until time $t$, that is, $Y_{0 \le s \le t}$.

**Theorem 1.** *[Thm 2.1 (Bain & Crisan, 2009)] Consider the probability triplet $(\Omega, \mathcal{F}, \mathrm{P})$, the metric space $\mathcal{S}$ and its Borel sigma-algebra $\mathcal{B}(\mathcal{S})$. There exists a (probability measure valued $\mathcal{P}(\mathcal{S})$) process $\{\pi_{0 \le t \le T}, \mathcal{F}_{0 \le t \le T}^Y\}$, with a progressively measurable modification, such that for all $A \in \mathcal{B}(\mathcal{S})$, the conditional probability measure $\mathrm{P}[X \in A \mid \mathcal{F}_t^Y]$ is well defined and is equal to $\pi_t(A)$.*

The conditional probability measure is extremely important, as the fundamental goal of nonlinear filtering is the solution of the following problem. Here, we introduce the quantity $\phi$, which is a random variable derived from the latent abstractions $X$.

**Problem 1.** *For any fixed $\phi : \mathcal{S} \to \mathbb{R}$ bounded and measurable, given knowledge of the measurement process $Y_{0 \le s \le t}$, compute $\mathbb{E}_{\mathrm{P}}[\phi(X) \mid \mathcal{F}_t^Y]$. This amounts to computing*

$$\langle \pi_t, \phi \rangle = \int_{\mathcal{S}} \phi(x) \mathrm{d}\pi_t(x). \tag{2}$$

In simple terms, Problem 1 involves studying the existence of the a-posteriori measure and the implementation of efficient algorithms for its update, using the flowing stream of incoming information $Y_t$. We first focus our attention on the existence of an analytic expression for the value of the a-posteriori expected measure $\pi_t$. Then, we quantify the interaction dynamics between observable measurements and $\phi$, through the lenses of mutual information $\mathcal{I}(Y_{0 \le s \le t}; \phi)$, which is an extension of the problems considered in (Newton, 2008; Duncan, 1970; 1971; Mitter & Newton, 2003).

## 2.1 TECHNICAL PRELIMINARIES

We set the stage of our work by revisiting the measurement process $Y_t$, and express it in a way that does not require access to unobservable information. Indeed, while $Y_t$ is naturally adapted w.r.t. its own filtration $\mathcal{F}_t^Y$, and consequently to any other growing filtration $\mathcal{R}_t$ such $\mathcal{F}_t^{Y,X} \supseteq \mathcal{R}_t \supseteq \mathcal{F}_t^Y$, the representation in Equation (1) is in general not adapted, letting aside degenerate cases.

Let's consider the family of growing filtrations $\mathcal{R}_t = \sigma(\mathcal{R}_0 \cup \sigma(Y_{0 \le s \le t} - Y_0))$, where $\sigma(Y_0) \subseteq \mathcal{R}_0 \subseteq \sigma(X, Y_0)$. Intuitively $\mathcal{R}_0$ allows to modulate between the two extreme cases of knowing only the initial conditions of the SDE, that is $Y_0$, to the case of complete knowledge of the whole latent abstraction $X$, and anything in between. As shown hereafter, the original process $Y_t$ associated to the space $(\Omega, \mathcal{F}, \mathrm{P})$ which solves Equation (1), also solves Equation (4), that is adapted on the reduced filtration $\mathcal{R}_t$. This allows us to reason about the partial observation of the latent abstraction ($\mathcal{R}_0$ vs $\sigma(X, Y_0)$), without incurring in the problem of the measurement process $Y_t$ being statistically dependent of the whole latent abstraction $X$.

Armed with such representation, we study under which change of measure the process $Y_t - Y_0$ behaves as a Brownian motion (Theorem 3). This serves the purpose of simplifying the calculation of the expected value of $\phi$ given $Y_t$, as described in Problem 1. Indeed, if $Y_t - Y_0$ is a Brownian motion independent of $\phi$, its knowledge does not influence our best guess for $\phi$, i.e. the conditional expected value. Moreover, our alternative representation is instrumental for the efficient and simple computation of the mutual information $\mathcal{I}(Y_{0 \le s \le t}; \phi)$, where the different measures involved in the Radon-Nikodym derivatives will be compared against the same reference Brownian measures.

The first step to define our representation is provided by the following

**Theorem 2.** *[Proof]. Consider the the probability triplet $(\Omega, \mathcal{F}, \mathrm{P})$, the process in Equation (1) defined on it, and the growing filtration $\mathcal{R}_t = \sigma(\mathcal{R}_0 \cup \sigma(Y_{0 \le s \le t} - Y_0))$. Define a new stochastic process*

$$W_t^{\mathcal{R}} \stackrel{def}{=} Y_t - Y_0 - \int_0^t \mathbb{E}_{\mathrm{P}}(H(Y_s, X, s) \mid \mathcal{R}_s) \mathrm{d}s. \tag{3}$$

*Then, $\{W_{0 \le t \le T}^{\mathcal{R}}, \mathcal{R}_{0 \le t \le T}\}$ is a Brownian motion. Notice that if $\mathcal{R}_t = \mathcal{F}_t^{Y,X}$, then $W_t^{\mathcal{R}} = W_t$.*

Following Theorem 2, the process $\{Y_{0 \leq t \leq T}, \mathcal{R}_{0 \leq t \leq T}\}$ has SDE representation

$$Y_t = Y_0 + \int_0^t \mathbb{E}_{\mathrm{P}}(H(Y_s, X, s) \,|\, \mathcal{R}_s)\mathrm{d}s + W_t^{\mathcal{R}}. \tag{4}$$

Next, we derive the change of measure necessary for the process $\tilde{W}_t \stackrel{\text{def}}{=} Y_t - Y_0$ to be a Brownian motion w.r.t to the filtration $\mathcal{R}_t$. To do this, we apply the Girsanov theorem (Øksendal, 2003) to $\tilde{W}_t$ which, in general, admits a $\mathcal{R}$ – adapted representation $\int_0^t \mathbb{E}_{\mathrm{P}}(H(Y_s, X, s) \,|\, \mathcal{R}_s)\mathrm{d}s + W_t^{\mathcal{R}}$.

**Theorem 3.** *[Proof]. Define the new probability space* $(\Omega, \mathcal{R}_T, \mathrm{Q}^{\mathcal{R}})$ *via the measure* $\mathrm{Q}^{\mathcal{R}}(A) = \mathbb{E}_{\mathrm{P}}\left[\mathbf{1}(A)(\psi_T^{\mathcal{R}})^{-1}\right]$, *for* $A \in \mathcal{R}_T$, *where*

$$\psi_t^{\mathcal{R}} \stackrel{\text{def}}{=} \exp\left( \int_0^t \mathbb{E}_{\mathrm{P}}[H(Y_s, X, s) \,|\, \mathcal{R}_s]\mathrm{d}Y_s - \frac{1}{2}\int_0^t \|\mathbb{E}_{\mathrm{P}}[H(Y_s, X, s) \,|\, \mathcal{R}_s]\|^2\mathrm{d}s \right), \tag{5}$$

*and*

$$\mathrm{Q}^{\mathcal{R}} \,|_{\mathcal{R}_t} = \mathbb{E}_{\mathrm{P}}\left[\mathbf{1}(A)\mathbb{E}_{\mathrm{P}}[(\psi_T^{\mathcal{R}})^{-1} \,|\, \mathcal{R}_t]\right] = \mathbb{E}_{\mathrm{P}}\left[\mathbf{1}(A)(\psi_t^{\mathcal{R}})^{-1}\right].$$

*Then, the stochastic process* $\{\tilde{W}_{0 \leq t \leq T}, \mathcal{R}_{0 \leq t \leq T}\}$ *is a Brownian motion on the space* $(\Omega, \mathcal{R}_T, \mathrm{Q}^{\mathcal{R}})$.

A direct consequence of Theorem 3 is that the process $\tilde{W}_t$ is independent of any $\mathcal{R}_0$ measurable random variable under the measure $\mathrm{Q}^{\mathcal{R}}$. Moreover, it holds that for all $\mathcal{R}_t' \subseteq \mathcal{R}_t$, $\mathrm{Q}^{\mathcal{R}} \,|_{\mathcal{R}_t'} = \mathrm{Q}^{\mathcal{R}'} \,|_{\mathcal{R}_t'}$.

## 2.2 A-Posteriori Measure and Mutual Information

As we did in § 2 for the process $\pi_t$, here we introduce a new process $\pi_t^{\mathcal{R}}$ which represents the conditional law of $X$ given the filtration $\mathcal{R}_t = \sigma(\mathcal{R}_0 \cup \sigma(Y_{0 \leq s \leq t} - Y_0))$. More precisely, for all $A \in \mathcal{B}(\mathcal{S})$, the conditional probability measure $\mathrm{P}[X \in A \,|\, \mathcal{R}_t]$ is well defined and is equal to $\pi_t^{\mathcal{R}}(A)$. Moreover, for any $\phi : \mathcal{S} \to \mathbb{R}$ bounded and measurable, $\mathbb{E}_{\mathrm{P}}[\phi(X) \,|\, \mathcal{R}_t] = \langle \pi_t^{\mathcal{R}}, \phi \rangle$. Notice that if $\mathcal{R} = \mathcal{F}^Y$ then $\pi^{\mathcal{R}}$ reduces to $\pi$.

Armed with Theorem 3, we are ready to derive the expression for the a-posteriori measure $\pi_t^{\mathcal{R}}$ and the mutual information between observable measurements and the unavailable information about the latent abstractions, that materialize in the random variable $\phi$.

**Theorem 4.** *[Proof]. The measure-valued process* $\pi_t^{\mathcal{R}}$ *solves in weak sense (see Appendix D for a precise definition), the following* SDE

$$\pi_t^{\mathcal{R}} = \pi_0^{\mathcal{R}} + \int_0^t \pi_s^{\mathcal{R}} \left(H(Y_s, \cdot, s) - \langle \pi_s^{\mathcal{R}}, H(Y_s, \cdot, s)\rangle\right)\left(\mathrm{d}Y_s - \langle \pi_s^{\mathcal{R}}, H(Y_s, \cdot, s)\rangle\mathrm{d}s\right), \tag{6}$$

*where the initial condition* $\pi_0$ *satisfies* $\pi_0^{\mathcal{R}}(A) = \mathrm{P}[X \in A \,|\, \mathcal{R}_0]$ *for all* $A \in \mathcal{B}(\mathcal{S})$.

When $\mathcal{R} = \mathcal{F}^Y$, Equation (6) is the well-know Kushner-Stratonovitch (or Fujisaki-Kallianpur-Kunita) equation (see e.g. Bain & Crisan (2009)). A proof for uniqueness of the solution of Equation (6) can be approached by considering the strategies in (Fotsa-Mbogne & Pardoux, 2017), but is outside the scope of this work. The (recursive) expression in Equation (6) is particularly useful for engineering purposes since, in general, it is usually not known in which variables $\phi(X)$, representing latent abstractions, we could be interested in. Keeping track of the *whole distribution* $\pi_t^{\mathcal{R}}$ at time $t$ is the most cost-effective solution, as we will show later.

Our next goal is to quantify the interaction dynamics between observable measurements and latent abstractions that materialize through the variable $\phi(X)$ (from now on we write only $\phi$ for the sake of brevity): in Theorem 5 we derive the mutual information $\mathcal{I}(Y_{0 \leq s \leq t}; \phi)$.

**Theorem 5.** *[Proof] The mutual information between observable measurements* $Y_{0 \leq s \leq t}$ *and* $\phi$ *is defined as:*

$$\mathcal{I}(Y_{0 \leq s \leq t}; \phi) \stackrel{\text{def}}{=} \int \log \frac{\mathrm{dP}_{\#Y_{0 \leq s \leq t}, \phi}}{\mathrm{dP}_{\#Y_{0 \leq s \leq t}}\mathrm{dP}_{\#\phi}}\mathrm{dP}_{\#Y_{0 \leq s \leq t}, \phi}. \tag{7}$$

*It holds that such quantity is equal to* $\mathbb{E}_{\mathrm{P}}\left[\log \frac{\mathrm{dP} \,|_{\mathcal{R}_t}}{\mathrm{dP} \,|_{\mathcal{F}_t^Y}\mathrm{dP} \,|_{\sigma(\phi)}}\right]$, *which can be simplified as follows:*

$$\mathcal{I}(Y_0; \phi) + \frac{1}{2}\mathbb{E}_{\mathrm{P}}\left[\int_0^t \|\mathbb{E}_{\mathrm{P}}[H(X, Y_s, s) \,|\, \mathcal{F}_s^Y] - \mathbb{E}_{\mathrm{P}}[H(X, Y_s, s) \,|\, \mathcal{R}_s]\|^2\mathrm{d}s\right]. \tag{8}$$

The mutual information computed by Equation (8) is composed by two elements: first, the mutual information between the initial measurements $Y_0$ and $\phi$, which is typically zero by construction. The second term quantifies how much the best prediction of the observation function $H$ is influenced by the extra knowledge of $\phi$, in addition to the measurement history $Y_{0 \leq s \leq t}$. By adhering to the premise that the conditional expectation of a stochastic variable constitutes the optimal estimator given the conditioning information, the integral on the r.h.s quantifies the expected square difference between predictions, having access to measurements only ($\mathbb{E}_{\mathrm{P}}[\cdot \,|\, \mathcal{F}_t^Y]$) and those incorporating additional information ($\mathbb{E}_{\mathrm{P}}[\cdot \,|\, \mathcal{R}_t]$).

Even though a precise characterization for general observation functions and and variables $\phi$ is typically out of reach, a **qualitative** analysis is possible. First, the mutual information between $\phi$ and the measurements depends on *i)* how much the amplitude of $H$ is impacted by knowledge of $\phi$ and *ii)* the *number* of elements of $H$ which are impacted (informally, how much localized vs global is the impact of $\phi$). Second, it is possible to define a hierarchical interpretation about the emergence of the various latent factors: a variable with a local impact can "*appear*", in an information theoretic sense, only if the impact of other global variables is resolved, otherwise the remaining uncertainty of the global variables makes knowledge of the local variable irrelevant. In classical diffusion models, this is empirically known (Chen et al., 2023; Linhardt et al., 2024; Tang et al., 2023), and corresponds to the phenomenon where *semantics emerges before details* (global vs local details in our language).

Now, consider any $\mathcal{F}_t^Y$ measurable random variable $\tilde{Y}_t$, defined as a mapping to a generic measurable space $(\Psi, \mathcal{B}(\Psi))$, which means it can also be seen as a process. The *data processing inequality* states that the mutual information between such $\tilde{Y}$ and $\phi$ will be smaller than the mutual information between the original measurement process and $\phi$. However, it can be shown that all the relevant information about the random variable $\phi$ contained in $\mathcal{F}_t^Y$ is equivalently contained in the filtering process at time instant $t$, that is $\pi_t$. This is not trivial, since $\pi_t$ is a $\mathcal{F}_t^Y$-measurable quantity, i.e., $\sigma(\pi_t) \subset \mathcal{F}_t^Y$. In other words, we show that $\pi_t$ is a **sufficient statistic** for any $\sigma(X)$ measurable random variable when starting from the measurement process.

**Theorem 6.** *[Proof] For any $\mathcal{F}_t^Y$ measurable random variable $\tilde{Y}_t : \Omega \to \Psi$, the following inequality holds:*

$$\mathcal{I}(\tilde{Y}; \phi) \leq \mathcal{I}(Y_{0 \leq s \leq t}; \phi). \tag{9}$$

*For a given $t \geq 0$, the measurement process $Y_{0 \leq s \leq t}$ and $X$ are conditionally-independent given $\pi_t$. This implies that $\mathrm{P}(A \,|\, \sigma(\pi_t)) = \mathrm{P}(A \,|\, \mathcal{F}_t^Y), \quad \forall A \in \sigma(X)$. Then $\mathcal{I}(Y_{0 \leq s \leq t}; \phi) = \mathcal{I}(\pi_t; \phi)$ (i.e. Equation (9) is attained with equality).*

While $\pi_t$ contains all the relevant information about $\phi$, the same cannot be said about the conditional expectation, i.e. the particular case $\tilde{Y} = \langle \pi_t, \phi \rangle$. Indeed, from Equation (2), $\langle \pi_t, \phi \rangle$ is obtained as a *transformation* of $\pi_t$ and thus can be interpreted as a $\mathcal{F}_t^Y$ measurable quantity subject to the constraint of Equation (9). As a particular case, the quantity $\langle \pi_t, H \rangle$, of central importance in the construction of generative models § 3, carries in general less information about $\phi$ than the un-projected $\pi_t$.

## 3 GENERATIVE MODELLING

We are interested in **generative models** for a given $\sigma(X)$-measurable random variable $V$.

An intuitive illustration of how data generation works according to our framework is as follows. Consider, for example, the image domain, and the availability of a rendering engine that takes as an input a computer program describing a scene (coordinates of objects, textures, light sources, auxiliary labels, etc ...) and that produces an output image of the scene. In a similar vein, a generative model learns how to use latent variables (which are not explicitly provided in input, but rather implicitly learned through training) to generate an image. For such model to work, one valid strategy is to consider an SDE in the form of Equation (1) where the following holds[1].

**Assumption 1.** *The stochastic process $Y_t$ satisfies $Y_T = V, \quad \mathrm{P} - a.s.$*

Then, we could numerically simulate the dynamics of Equation (1) until time $T$. Indeed, starting from initial conditions $Y_0$, we could obtain $Y_T$ that, under Assumption 1, is precisely $V$. Unfortunately,

---

[1]From a strict technical point of view, Assumption 1 might be incompatible with other assumptions in Appendix A, or proving compatibility could require particular effort. Such details are discussed in Appendix G.

such a simple idea requires *explicit access* to $X$, as it is evident from Equation (1). In mathematical terms, Equation (1) is adapted to the filtration $\mathcal{F}_t^{Y,X}$. However, we have shown how to reduce the available information to account only for historical values of $Y_t$. Then, we can combine the result in Theorem 4 with Theorem 2 and re-interpret Equation (4), which is a valid generative model, as

$$\begin{cases} \pi_t = \pi_0 + \int_0^t \pi_s \left( H - \langle \pi_s, H \rangle \right) \left( \mathrm{d}Y_s - \langle \pi_s, H \rangle \mathrm{d}s \right), \\ Y_t = Y_0 + \int_0^t \langle \pi_s, H \rangle \mathrm{d}s + W_t^{\mathcal{F}^Y}, \end{cases} \tag{10}$$

where $H$ denotes $H(Y_s, \cdot, s)$. Explicit simulation of Equation (10) only requires knowledge of the whole history of the measurement process: provided Assumption 1 holds, it allows generation of a sample of the random variable $V$.

Although the discussion in this work includes a large class of observation functions, we focus on the particular case of generative diffusion models (Song et al., 2021). Typically, such models are presented through the lenses of a forward noising process and backward (in time) SDEs, following the intuition of Anderson (1982). Next, according to the framework we introduce in this work, we reinterpret such models under the perspective of enlargement of filtrations.

Consider the *reversed* process $\hat{Y}_t \overset{\text{def}}{=} Y_{T-t}$ defined on $(\Omega, \mathcal{F}, \mathrm{P})$ and the corresponding filtration $\mathcal{F}_t^{\hat{Y}} \overset{\text{def}}{=} \sigma(\hat{Y}_{0 \leq s \leq t})$. The measure P is selected such that the process $\hat{Y}_t$ has $\mathcal{F}_t^{\hat{Y}}$–adapted expression

$$\hat{Y}_t = V + \int_0^t F(\hat{Y}_s, s) \mathrm{d}s + \hat{W}_t, \tag{11}$$

where $\{\hat{W}_t, \mathcal{F}_t^{\hat{Y}}\}$ is a Brownian motion. Then, Assumption 1 is valid since $Y_T = \hat{Y}_0 = V$. Note that Equation (11), albeit with a different notation, is reminiscent of the forward SDE that is typically used as the starting point to illustrate score-based generative models (Song et al., 2021). In particular, $F(\cdot)$ corresponds to the drift term of such a diffusion SDE.

Equation (11) is equivalent to $Y_t = V + \int_t^T F(Y_s, T - s) \mathrm{d}s + \hat{W}_{T-t}$, which is an expression for the process $Y_t$, which is adapted to $\mathcal{F}^{\hat{Y}}$. This constitutes the first step to derive an equivalent backward (generative) process according to the traditional framework of score-based diffusion models. Note that such an equivalent representation is not useful for simulation purposes: the goals of the next steps is to transform it such that it is adapted to $\mathcal{F}^Y$. Indeed, using simple algebra, it holds that

$$Y_t = Y_0 - \int_0^t F(Y_s, T - s) \mathrm{d}s + \left( -Y_0 + V + \int_0^T F(Y_s, T - s) \mathrm{d}s + \hat{W}_{T-t} \right),$$

where the last term in the parentheses is equal to $-\hat{W}_T + \hat{W}_{T-t}$.

Note that $\mathcal{F}_t^Y = \sigma(\hat{Y}_{T-t \leq s \leq T})$. Since $\sigma(\hat{Y}_{T-t \leq s \leq T}) = \sigma(\hat{W}_{T-t \leq s \leq T}) \cup \sigma(\hat{Y}_{T-t})$, we can apply the result in (Pardoux, 2006) (Thm 2.2) to claim the following: $-\hat{W}_T + \hat{W}_{T-t} - \int_0^t \nabla \log \hat{p}(Y_s, T - s) \mathrm{d}s$ is a Brownian motion adapted to $\mathcal{F}_t^Y$, where this time $\mathrm{P}(\hat{Y}_t \in \mathrm{d}y) = \hat{p}(y, t) \mathrm{d}y$. Then (Pardoux, 2006)

**Theorem 7.** *Consider the stochastic process $Y_t$ which solves Equation* (11). *The same stochastic process also admits a $\mathcal{F}_t^Y$–adapted representation*

$$Y_t = Y_0 + \int_0^t \underbrace{-F(Y_s, T - s) + \nabla \log \hat{p}(Y_s, T - s)}_{\textit{In Theorem 8, we call this } F'(Y_s, s)} \mathrm{d}s + W_t. \tag{12}$$

Equation (12) corresponds to the backward diffusion process from (Song et al., 2021) and, because it is adapted to the filtration $\mathcal{F}^Y$, it represents a valid, and easy to simulate, measurement process.

By now, it is clear how to go from an $\mathcal{F}^{Y,X}$–adapted filtration to a $\mathcal{F}^Y$–adapted one. We also showed that a $\mathcal{F}^Y$–adapted filtration can be linked to the reverse, $\mathcal{F}^{\hat{Y}}$–adapted process induced by a forward

diffusion SDE. What remains to be discussed is the connection that exists between the $\mathcal{F}^Y$–adapted filtration, and its *enlarged* version $\mathcal{F}^{Y,X}$. In other words, we have shown that a forward, diffusion SDE admits a backward process which is compatible with our generative model that simulates a NLF process having access only to measurements, but we need to make sure that such process admits a formulation that is compatible the standard NLF framework in which latent abstractions are available.

To do this, we can leverage existing results about Markovian bridges (Rogers & Williams, 2000; Ye et al., 2022) (and further work (Aksamit et al., 2017; Ouwehand, 2022; Grigorian & Jarrow, 2023; Çetin & Danilova, 2016) on filtration enlargement). This requires assumptions about the existence and well-behavedness of densities $p(y, t)$ of the SDE process, defined by the logarithm of the Radon-Nikodym derivative of the instantaneous measure $\mathrm{P}(Y_t \in \mathrm{d}y)$ w.r.t. the Lebesgue measure in $\mathbb{R}^N$, $\mathrm{P}(Y_t \in \mathrm{d}y) = p(y, t)\mathrm{d}y^2$.

**Theorem 8.** *Suppose that on $(\Omega, \mathcal{F}, \mathrm{P})$ the Markov stochastic process $Y_t$ satisfies*

$$Y_t = Y_0 + \int_0^t F'(Y_s, s)\mathrm{d}s + W_t, \tag{13}$$

*where $\{W_{0 \leq t \leq T}, \mathcal{F}^Y_{0 \leq t \leq T}\}$ is a Brownian motion and $F$ satisfies the requirements for existence and well definition of the stochastic integral (Shreve, 2004). Moreover, let Assumption 1 hold. Then, the same process admits $\mathcal{R}_t = \sigma(Y_{0 \leq s \leq t}, Y_T)$–adapted representation*

$$Y_t = Y_0 + \int_0^t F'(Y_s, s) + \nabla_{Y_s} \log p(Y_T \,|\, Y_s)\mathrm{d}s + \beta_t, \tag{14}$$

*where $p(Y_T \,|\, Y_s)$ is the density w.r.t the Lebesgue measure of the probability $\mathrm{P}(Y_T \,|\, \sigma(Y_s))$, and $\{\beta_{0 \leq t \leq T}, \mathcal{R}_{0 \leq t \leq T}\}$ is a Brownian motion.*

The connection between time reversal of diffusion processes and enlarged filtrations is finalized with the result of Al-Hussaini & Elliott (1987), Thm. 3.3, where it is proved how the $\beta_t$ term of Equation (14) is a Brownian motion, using the techniques of time reversals of SDEs.

Since $\hat{p}(y, T - t) = p(y, t)$, the enlarged filtration version of Equation (12) reads

$$Y_t = Y_0 + \int_0^t \underbrace{-F(Y_s, T - s) + \nabla_{Y_s} \log p(Y_s \,|\, Y_T)\mathrm{d}s}_{\text{Equivalent to } H(Y_t, X, t) = -F(Y_s, T-s) + \nabla_{Y_s} \log p(Y_s \,|\, g(X))} + W_t. \tag{15}$$

Note that the dependence of $Y_t$ on the latent abstractions $X$ is implicitly defined by conditioning the score term $\nabla_{Y_s} \log p(Y_s \,|\, Y_T)$ by $Y_T$, which is the "rendering" of $X$ into the observable data domain.

Clearly, Equation (15) can be reverted to the starting generative Equation (12) by mimicking the results which allowed us to go from Equation (1) to Equation (4), by noticing that $\mathbb{E}_\mathrm{P}[\nabla_{Y_s} \log p(Y_T \,|\, Y_s) \,|\, \mathcal{F}^Y_t] = 0$ (informally, this is obtained since $\int \nabla_{y_s} \log p(y_t \,|\, y_s) p(y_t \,|\, y_s)\mathrm{d}y_t = \int \nabla_{y_s} p(y_t \,|\, y_s)\mathrm{d}y_t = 0$).

It is also important to notice that we can derive the expression for the mutual information between the measurement process and a sample from the data distribution, as follows

$$\mathcal{I}(Y_{0 \leq s \leq t}; V) = \mathcal{I}(Y_0; V) + \frac{1}{2}\mathbb{E}_\mathrm{P}\left[\int_0^t \|\nabla_{Y_s} \log p(Y_s) - \nabla_{Y_s} \log p(Y_s \,|\, Y_T)\|^2 \mathrm{d}s\right]. \tag{16}$$

Mutual information is tightly related to the classical loss function of generative diffusion models.

Furthermore, by casting the result of Equation (8) according to the forms of Equations (12) and (15), we obtain the simple and elegant expression

$$\mathcal{I}(Y_{0 \leq s \leq t}; V) = \mathcal{I}(Y_0; V) + \frac{1}{2}\mathbb{E}_\mathrm{P}\left[\int_0^t \|\nabla_{Y_s} \log p(Y_T \,|\, Y_s)\|^2 \mathrm{d}s\right]. \tag{17}$$

In Appendix H, we present a specialization of our framework for the particular case of linear diffusion models, recovering the expressions for the variance-preserving and variance-exploding SDEs that are the foundations of score-based generative models (Song et al., 2021).

---

[2]Similarly to what discussed in footnote 1, the analysis of the existence of the process adapted to $\mathcal{F}^Y_t$ is considered in the time interval $[0, T)$ (Haussmann & Pardoux, 1986). See also Appendix G.

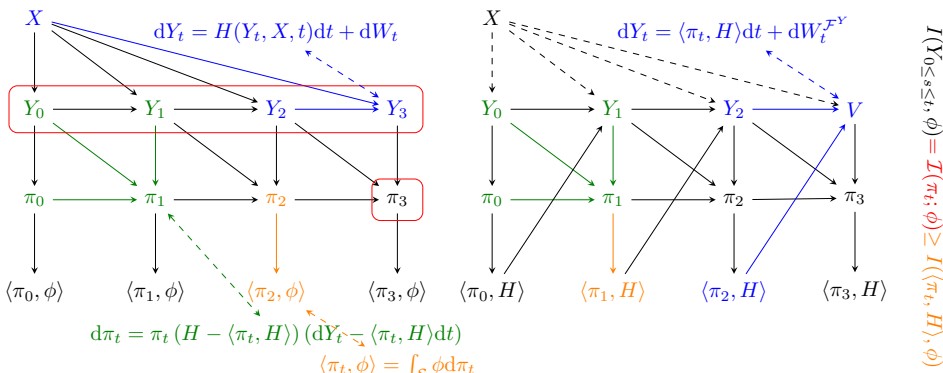

**Figure 1:** Graphical intuition for our results: nonlinear filtering (left) and generative modelling (right).

## 4 AN INFORMAL SUMMARY OF THE RESULTS

We shall now take a step back from the rigour of this work, and provide an intuitive summary of our results, using Figure 1 as a reference. We begin with an illustration of NLF, shown on the left of the figure. We consider an observable latent abstraction $X$ and the measurement process $Y_t$, which for ease of illustration we consider evolving in discrete time, i.e. $Y_0, Y_1, \ldots$, and whose joint evolution is described by Equation (1). Such interaction is shown in blue: $Y_3$ depends on its immediate past $Y_2$ and the latent abstraction $X$.

The a-posteriori measure process $\pi_t$ is updated in an iterative fashion, by integrating the flux of information. We show this in green: $\pi_1$ is obtained by updating $\pi_0$ with $Y_1 - Y_0$ (the equivalent of $\mathrm{d}Y_t$). This evolution is described by Kushner's equation, which has been derived informally from the result of Equation (6). The a-posteriori process is a sufficient statistic for the latent abstraction $X$: for example, $\pi_3$ contains the same information about $\phi$ as the whole $Y_0, \ldots, Y_3$ (red boxes). Instead, in general, a projected statistic $\langle \pi_t, \phi \rangle$ contains less information than the whole measurement process (this is shown in orange, for time instant 2). The mutual information between all these variables is proven in Theorem 6, whereas the actual value of $\mathcal{I}(Y_{0 \leq s \leq t}; \phi)$ is shown in Theorem 5.

Next, we focus on generative modelling. As by our definition, any stochastic process satisfying Assumption 1 ($Y_3 = V$, in the figure) can be used for generative purposes. Since the latent abstraction is by definition not available, it is not possible to simulate directly the dynamics using Equation (1) (dashed lines from $X$ to $Y_t$). Instead, we derive a version of the process adapted to the history of $Y_t$ alone, together with the update of the projection $\langle \pi_t, H \rangle$, which amounts to simulating Equation (10).

The update of the upper part of Equation (10), which is a particular case of Equation (6), can be **interpreted** as the composition of two steps: 1) (green) the update of the a-posteriori measure given new available measurements, and, 2) (orange) the projection of the whole $\pi_t$ into the statistic of interest. The update of the measurement process, i.e. the lower part of Equation (10), is color-coded in blue. This is in stark contrast to the NLF case, as the update of e.g. $Y_3 = V$ does not depend **directly** on $X$. The system in Equation (10) and its simulation describes the emergence of latent world representations in SDE-based generative models:

> We interpret the $\mathcal{F}_t^Y$ measurable quantity $\langle \pi_t, H \rangle$ as the cascade of mappings trough the spaces
>
> $$\langle \pi_t, H \rangle : \quad \mathcal{C}([0, t], \mathbb{R}^N) \to \mathcal{P}(\mathcal{S}) \times \mathbb{R}^N \to \mathbb{R}^N$$
> $$Y_{0 \leq s \leq t} \to (\pi_t, Y_t) \to \langle \pi_t, H \rangle$$
>
> We consider it as a mapping that **first** transforms the whole $Y_{0 \leq s \leq t}$ into the *condensed* (in terms of sufficient statistics Theorem 6) $\pi_t$, keep also $Y_t$, and **second** uses these two to construct $\langle \pi_t, H \rangle$.

The theory developed in this work guarantees that the mutual information between measurements and any statistics $\phi$, grows as described by Theorem 5. Our framework offers a new perspective, according to which, the dynamics of SDE-based generative models (Song et al., 2021) implicitly mimic the two steps procedure described in the box above. We claim that this is the reason why

it is possible to dissect the parametric drift of such generative models and find a *representation* of the abstract state distribution $\pi_t$, encoded into their activations. Next, we set to root our theoretical findings in experimental evidence.

## 5 EMPIRICAL EVIDENCE

We complement existing empirical studies (Park et al., 2023; Kwon et al., 2023; Chen et al., 2023; Linhardt et al., 2024; Tang et al., 2023; Xiang et al., 2023; Haas et al., 2024; Sclocchi et al., 2024) that first measured the interactions between the generative process of diffusion models and latent abstractions, by focusing on a particular dataset that allows for a fine grained assessment of the influence of latent factors.

**Dataset.** We use the Shapes3D (Kim & Mnih, 2018) dataset, which is a collection of $64 \times 64$ ray-tracing generated images, depicting simple 3D-scenes, with an object (a sphere, cube, ...) placed in a space, described by several attributes (color, size, orientation). Attributes have been derived from the computer program that the ray-tracing software executed to generate the scene: these are transformed into labels associated to each image. In our experiments, such labels are the materialization of the latent abstractions $X$ we consider in this work (see Appendix J.1 for details).

**Measurement Protocols.** For our experiments, we use the base NCSPP model described by Song et al. (2021): specifically, our denoising score network corresponds to a U-NET (Ronneberger et al., 2015). We train the unconditional version of this model from scratch, using score-matching objective. Detailed hyper-parameters and training settings are provided in Appendix J.2. Next, we summarize three techniques to measure the emergence of latent abstractions through the lenses of the labels associated to each image in our dataset. For all such techniques, we use a specific "measurement" subset of our dataset, which we partition in 246 training, 150 validation, and 371 test examples. We use a multi-label stratification algorithm (Sechidis et al., 2011; Szymański & Kajdanowicz, 2017) to guarantee a balanced distribution of labels across all dataset splits.

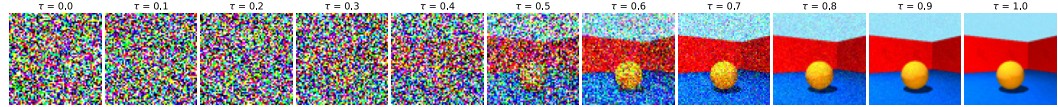

**Figure 2:** Versions of an image corrupted by different values of noise for different times $\tau$.

*Linear probing*. Each image in the measurement subset is perturbed with noise, using a variance-exploding schedule (Song et al., 2021), with noise levels decreasing from $\tau = 0$ to $\tau = 1.0$ in steps of 0.1, as shown in Figure 2. Intuitively, each time value $\tau$ can be linked to a different Signal to Noise Ratio ($SNR$), ranging from $SNR(\tau = 1) = \infty$ to $SNR(\tau = 0) \simeq 0$. We extract several feature maps from all the linear and convolutional layers of the denoising score network, for each perturbed image, resulting in a total of 162 feature map sets for each noise level. This process yields 11 different datasets per layer, which we use to train a linear classifier (our probe) for each of these datasets, using the training subset. In these experiments, we use a batch size of 64 and adjust the learning rate based on the noise level (see Appendix J.3). Classifier performance is optimized by selecting models based on their log-probability accuracy observed on the validation subset. The final evaluation of each classifier is conducted on the test subset. Classification accuracy, measured by the model log likelihood, is a proxy of latent abstraction emergence (Chen et al., 2023).

*Mutual information estimation*. We estimate mutual information between the labels and the outputs of the diffusion model across varying diffusion times, using Equation (39) (which is a specialized version of our theory for linear diffusion models, see Appendix H) and adopt the same methodology discussed by Franzese et al. (2024) to learn conditional and unconditional score functions, and to approximate the mutual information. The training process uses a randomized conditioning scheme: 33% of training instances are conditioned on all labels, 33% on a single label, and the remaining 33% are trained unconditionally. See Appendix J.4 for additional details.

*Forking*. We propose a new technique to measure at which stage of the generative process, image features described by our labels emerge. Given an initial noise sample, we proceed with numerical integration of the backward SDE (Song et al., 2021) up to time $\tau$. At this point, we fork $k$ replicas

of the backward process, and continue the $k$ generative pathways independently until numerical integration concludes. We use a simple classifier (a pre-trained ResNet50 (He et al., 2016) with an additional linear layer trained from scratch) to verify that labels are coherent across the $k$ forks. Coherency is measured using the entropy of the label distribution output by our simple classifier on each latent factor for all the $k$ forks. Intuitively: if we fork the process at time $\tau = 0.6$, and the $k$ forks all end up displaying a cube in the image (entropy equals 0), this implies that the object shape is a latent abstraction that has already emerged by time $\tau$. Conversely, lack of coherence implies that such a latent factor has not yet influenced the generative process. Details of the classifier training and sampling procedure are provided in Appendix J.5.

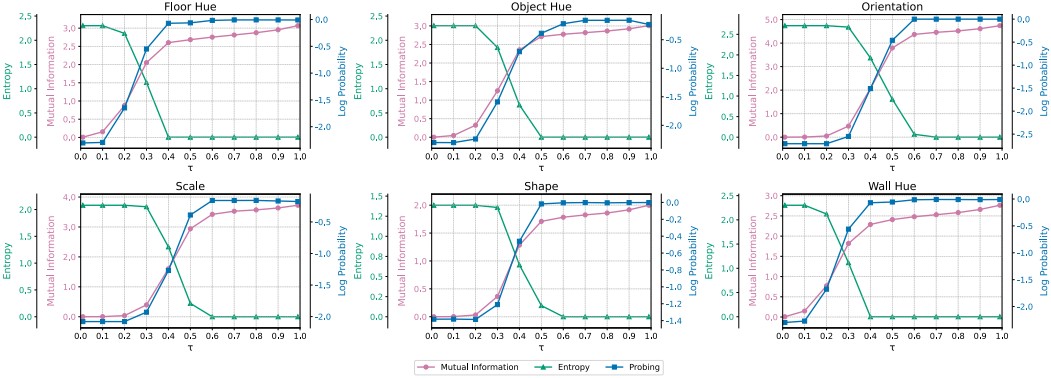

**Figure 3:** Mutual information, Entropy across forked generative pathways, and Probing results as functions of $\tau$.

**Results.** We present our results in Figure 3. We note that some attributes like *floor hue*, *wall hue* and *shape* emerge earlier than others, which corroborates the hierarchical nature of latent abstractions, a phenomenon that is related to the spatial extent of each attribute in pixel space. This is evident from the results of linear probing, where we evaluate the performance of linear probes trained on features maps extracted from the denoiser network, and from the mutual information measurement strategy and the measured entropy of the predicted labels across forked generative pathways. Entropy decreases with $\tau$, which marks the moment in which the generative process proceeds along $k$ forks. When generative pathways converge to a unique scene with identical predicted labels (entropy reaches zero), this means that the model has committed to a specific set of latent factors. This coincides with the same noise level corresponding to high accuracy for the linear probe, and high-values of mutual information. Further ablation experiments are presented in Appendix J.6.

## 6 CONCLUSION

Despite their tremendous success in many practical applications, a deep understanding of how SDE-based generative models operate remained elusive. A particularly intriguing aspect of several empirical work was to uncover the capacity of generative models to create entirely new data by combining latent factors learned from examples. To the best of our knowledge, there exist no theoretical framework that attempted at describing such phenomenon.

In this work, we closed this gap, and presented a novel theory — that builds on the framework of NLF — to describe the implicit dynamics allowing SDE-based generative models to tap into latent abstractions and guide the generative process. Our theory, that required advancing the standard NLF formulation, culminates in a new system of joint SDEs that fully describe the iterative process of data generation. Furthermore, we derived an information-theoretic measure to study the influence of latent abstractions, which provides a concrete understanding of the joint dynamics.

To root our theory into concrete examples, we collected experimental evidence by means of novel (and established) measurement strategies, that corroborates our understanding of diffusion models. Latent abstractions emerge according to an implicitly learned hierarchy, and can appear early on in the data generation process, much earlier than what is visible in the data domain. Our theory is especially useful as it allows analyses and measurements of generative pathways, opening up opportunities for a variety of applications, including image editing, and improved conditional generation.

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

# A ASSUMPTIONS

**Assumption 2.** *Whenever we mention a filtration, we assume as usual that it is augmented with the* $\mathrm{P}-$ *null sets, i.e. if the set $N$ is such that $\mathrm{P}(N) = 0$, then all $A \subseteq N$ should be in the filtration.*

**Assumption 3.**

$$\mathbb{E}_{\mathrm{P}}\big[\int_0^t \|H(Y_s, X, s)\|\mathrm{d}s\big] < \infty. \tag{18}$$

**Assumption 4.**

$$\mathrm{P}\big(\int_0^t \big\|\mathbb{E}_{\mathrm{P}}[H(Y_s, X, s) \,|\, \mathcal{F}_s^Y]\big\|^2 \mathrm{d}s < \infty\big) = 1. \tag{19}$$

Eq 2.5 Fundamentals of Stochastic Filtering. Necessary for validity of Equation (3).

**Assumption 5.**

$$\mathbb{E}_{\mathrm{P}}\big[\int_0^t \|H(Y_s, X, s)\|^2 \mathrm{d}s\big] < \infty. \tag{20}$$

*Note: this assumption implies Assumption 3 and Assumption 4. Despite it is more restrictive, it turns out that it is often easier to check.*

Eq 3.19 Fundamentals of Stochastic Filtering. Necessary for validity of Theorem 3.

**Assumption 6.**

$$\mathbb{E}_{\mathrm{P}}[\exp\Big\{\frac{1}{2}\int_0^t \|H(Y_s, X, s)\|^2 \mathrm{d}s\Big\}] < \infty, \tag{21}$$

*and*

$$\mathbb{E}_{\mathrm{P}}[\exp\Big\{\frac{1}{2}\int_0^t \|\mathbb{E}_{\mathrm{P}}[H(Y_s, X, s) \,|\, \mathcal{R}_s]\|^2 \mathrm{d}s\Big\}] < \infty, \tag{22}$$

Note: Assumption 6, as well as Assumption 5, are trivially verified when $H$ is bounded.

# B PROOF OF THEOREM 2

We start by combining Equation (3) and Equation (1)

$$W_t^{\mathcal{R}} = Y_0 + \int_0^t H(Y_s, X, s)\mathrm{d}s + W_t - Y_0 - \int_0^t \mathbb{E}_{\mathrm{P}}(H(Y_s, X, s) \,|\, \mathcal{R}_s)\mathrm{d}s$$

$$= \int_0^t H(Y_s, X, s)\mathrm{d}s + W_t - \int_0^t \mathbb{E}_{\mathrm{P}}(H(Y_s, X, s) \,|\, \mathcal{R}_s)\mathrm{d}s.$$

We begin by showing that it is a martingale. For any $0 \leq \tau \leq t$ it holds

$$\mathbb{E}_{\mathrm{P}}[W_t^{\mathcal{R}} \,|\, \mathcal{R}_\tau] = \mathbb{E}_{\mathrm{P}}\big[\int_0^t H(Y_s, X, s)\mathrm{d}s \,|\, \mathcal{R}_\tau\big] + \mathbb{E}_{\mathrm{P}}[W_t \,|\, \mathcal{R}_\tau]$$

$$\qquad - \mathbb{E}_{\mathrm{P}}\big[\int_0^t \mathbb{E}_{\mathrm{P}}(H(s, Y_s, X) \,|\, \mathcal{R}_s)\mathrm{d}s \,|\, \mathcal{R}_\tau\big]$$

$$= \int_0^t \mathbb{E}_{\mathrm{P}}[H(Y_s, X, s) \,|\, \mathcal{R}_\tau]\mathrm{d}s + \mathbb{E}_{\mathrm{P}}[\mathbb{E}_{\mathrm{P}}[W_t \,|\, \mathcal{F}_\tau^{Y,X}] \,|\, \mathcal{R}_\tau]$$

$$\qquad - \int_0^\tau \mathbb{E}_{\mathrm{P}}[H(Y_s, X, s) \,|\, \mathcal{R}_s]\mathrm{d}s - \int_\tau^t \mathbb{E}_{\mathrm{P}}[H(Y_s, X, s) \,|\, \mathcal{R}_\tau]\mathrm{d}s$$

$$= \int_0^\tau \mathbb{E}_{\mathrm{P}}[H(Y_s, X, s) \,|\, \mathcal{R}_\tau]\mathrm{d}s + \mathbb{E}_{\mathrm{P}}[W_\tau \,|\, \mathcal{R}_\tau] + W_\tau^{\mathcal{R}} + Y_0 - Y_\tau$$

$$= \mathbb{E}_{\mathrm{P}}\big[\int_0^\tau H(Y_s, X, s)\mathrm{d}s + W_\tau + Y_0 - Y_\tau \,|\, \mathcal{R}_\tau\big] + W_\tau^{\mathcal{R}} = W_\tau^{\mathcal{R}}.$$

Moreover, it is easy to check that the cross-variation of $W_t^{\mathcal{R}}$ is the same as the one of $W_t$. Then, we can conclude the proof by Levy's characterization of Brownian motion ($W_0^{\mathcal{R}} = 0$).

## C    PROOF OF THEOREM 3

First, by combining the definition of $\psi_t^{\mathcal{R}}$ and the fact that $\mathrm{d}Y_t = \mathbb{E}_\mathrm{P}[H(Y_t, X, t) \mid \mathcal{R}_t] + \mathrm{d}W_t^{\mathcal{R}}$ we obtain

$$(\psi_t^{\mathcal{R}})^{-1} = \exp\left(-\int_0^t \mathbb{E}_\mathrm{P}[H(Y_s, X, s) \mid \mathcal{R}_s]\mathrm{d}W_s^{\mathcal{R}} - \frac{1}{2}\int_0^t \|\mathbb{E}_\mathrm{P}[H(Y_s, X, s) \mid \mathcal{R}_s]\|^2\mathrm{d}s\right). \quad (23)$$

Notice that by Assumption 6 (which is actually the usual Novikov's condition), the local martingale $(\psi_t^{\mathcal{R}})^{-1}$ is a real-valued martingale starting from $(\psi_0^{\mathcal{R}})^{-1} = 1$. Then, we can apply Girsanov theorem and conclude that $\mathrm{d}Q^{\mathcal{R}} = \psi_T^{\mathcal{R}}\mathrm{d}P$ is a probability measure under which the process $\{\tilde{W}_{0 \leq t \leq T}, \mathcal{R}_{0 \leq t \leq T}\}$, with

$$\tilde{W}_t = W_t^{\mathcal{R}} + \int_0^t \mathbb{E}_\mathrm{P}[H(Y_t, X, s) \mid \mathcal{R}_t]\mathrm{d}s,$$

is a Brownian motion on the space $(\Omega, \mathcal{R}_T, Q^{\mathcal{R}})$.

## D    PROOF OF THEOREM 4

First, let us give a precise meaning to being a weak solution of Equation (6). We say that $\pi_t^{\mathcal{R}}$ solves (6) in a weak sense in, for any for any $\phi : \mathcal{S} \to \mathbb{R}$ bounded and measurable, it holds

$$\langle \pi_t^{\mathcal{R}}, \phi \rangle = \langle \pi_0^{\mathcal{R}}, \phi \rangle$$
$$+ \int_0^t \left(\langle \pi_s^{\mathcal{R}}, H(Y_s, \cdot, s)\phi \rangle - \langle \pi_s^{\mathcal{R}}, \phi \rangle\langle \pi_s^{\mathcal{R}}, H(Y_s, \cdot, s)\rangle\right)\left(\mathrm{d}Y_s - \langle \pi_s^{\mathcal{R}}, H(Y_s, \cdot, s)\rangle\mathrm{d}s\right). \quad (24)$$

Let us recall that, on $(\Omega, \mathcal{F}, \mathrm{P})$, the process $Y_t$ has the SDE representation (1), where $\{W_{0 \leq t \leq T}, \mathcal{F}_{0 \leq t \leq T}^{Y,X}\}$ is a Brownian motion. Moreover, by Theorem 3 with $\mathcal{R}_t = \mathcal{F}_t^{Y,X}$, it holds that $\{(Y - Y_0)_{0 \leq t \leq T}, \mathcal{F}_{0 \leq t \leq T}^{Y,X}\}$ is a Brownian motion on the space $(\Omega, \mathcal{F}, Q^{\mathcal{F}^{Y,X}})$, where $\mathrm{d}Q^{\mathcal{F}^{Y,X}} = (\psi_T^{\mathcal{F}^{Y,X}})^{-1}\mathrm{d}P$ and

$$\psi_t^{\mathcal{F}^{Y,X}} = \exp\left(\int_0^t H(Y_s, X, s)\mathrm{d}Y_s - \frac{1}{2}\int_0^t \|H(Y_s, X, s)\|^2\mathrm{d}s\right). \quad (25)$$

For notation simplicity, in this subsection $\psi_t^{\mathcal{F}^{Y,X}}$ and $Q^{\mathcal{F}^{Y,X}}$ are simply indicated as $\pi_t$, $\psi_t$ and $Q$ respectively.

Since we aim at showing that (24) holds, let us fix $\phi$ and let us start from $\mathbb{E}_\mathrm{P}[\phi(X) \mid \mathcal{R}_t] = \langle \pi_t^{\mathcal{R}}, \phi \rangle$. Bayes Theorem provides us with the following

$$\langle \pi_t^{\mathcal{R}}, \phi \rangle = \mathbb{E}_\mathrm{P}[\phi(X) \mid \mathcal{R}_t] = \frac{\mathbb{E}_\mathrm{Q}[\frac{\mathrm{d}P}{\mathrm{d}Q}\phi(X) \mid \mathcal{R}_t]}{\mathbb{E}_\mathrm{Q}[\frac{\mathrm{d}P}{\mathrm{d}Q} \mid \mathcal{R}_t]} = \frac{\mathbb{E}_\mathrm{Q}[\psi_T\phi(X) \mid \mathcal{R}_t]}{\mathbb{E}_\mathrm{Q}[\psi_T \mid \mathcal{R}_t]} \stackrel{\text{def}}{=} \frac{\langle \hat{\pi}_t^{\mathcal{R}}, \phi \rangle}{\langle \hat{\pi}_t^{\mathcal{R}}, 1 \rangle}. \quad (26)$$

Starting from the numerator $\langle \hat{\pi}_t^{\mathcal{R}}, \phi \rangle$, we involve the tower property of conditional expectation and the fact that $\psi_t$ is $\mathcal{F}_t^{Y,X}$ measurable to write

$$\langle \hat{\pi}_t^{\mathcal{R}}, \phi \rangle = \mathbb{E}_\mathrm{Q}[\psi_T\phi(X) \mid \mathcal{R}_t] = \mathbb{E}_\mathrm{Q}\left[\mathbb{E}_\mathrm{Q}\left[\psi_T\phi(X) \mid \mathcal{F}_t^{Y,X}\right] \mid \mathcal{R}_t\right]$$
$$= \mathbb{E}_\mathrm{Q}\left[\mathbb{E}_\mathrm{Q}\left[\psi_T \mid \mathcal{F}_t^{Y,X}\right]\phi(X) \mid \mathcal{R}_t\right] = \mathbb{E}_\mathrm{Q}\left[\psi_t\phi(X) \mid \mathcal{R}_t\right]. \quad (27)$$

Recalling the definition of $\psi_t$ (see Equation (25)), we have

$$\mathrm{d}\psi_t = \psi_t H(Y_t, X, t)\mathrm{d}Y_t, \quad (28)$$

from which it follows

$$\psi_t = 1 + \int_0^t \psi_s H(Y_s, X, s)\mathrm{d}Y_s. \quad (29)$$

We continue processing Equation (27), using Equation (29), as

$$\mathbb{E}_Q\left[\psi_t\phi(X)\,|\,\mathcal{R}_s\right] = \mathbb{E}_Q\left[\left(1 + \int_0^t \psi_s H(Y_s, X, s)\mathrm{d}Y_s\right)\phi(X)\,|\,\mathcal{R}_t\right]$$

$$= \mathbb{E}_Q\left[\phi(X)\,|\,\mathcal{R}_t\right] + \mathbb{E}_Q\left[\int_0^t \psi_s H(Y_s, X, s)\phi(X)\mathrm{d}Y_s\,|\,\mathcal{R}_t\right]$$

$$= \mathbb{E}_Q\left[\phi(X)\,|\,\mathcal{R}_t\right] + \int_0^t \mathbb{E}_Q\left[\psi_s H(Y_s, X, s)\phi(X)\,|\,\mathcal{R}_s\right]\mathrm{d}Y_s,$$

where to obtain the last equality we used Lemma 5.4 in Xiong (2008). We also recall that, under Q, the process $(Y_t - Y_0)$ is independent of $X$. Thus, since $\mathcal{R}_t = \sigma(\mathcal{R}_0 \cup \sigma(Y_{0 \le s \le t} - Y_0))$ and $\frac{\mathrm{d}P}{\mathrm{d}Q}\,|\,_{\mathcal{F}_0^{Y,X}} = 1$, we obtain $\mathbb{E}_Q\left[\phi(X)\,|\,\mathcal{R}_t\right] = \mathbb{E}_P[\phi(X)\,|\,\mathcal{R}_0]$. Concluding and rearranging:

$$\langle\hat{\pi}_t^{\mathcal{R}}, \phi\rangle = \langle\hat{\pi}_0^{\mathcal{R}}, \phi\rangle + \int_0^t \langle\hat{\pi}_s^{\mathcal{R}}, \phi H(Y_s, \cdot, s)\rangle\mathrm{d}Y_s.$$

Obviously by the same arguments $\langle\hat{\pi}_t^{\mathcal{R}}, 1\rangle = \mathbb{E}_Q[\frac{\mathrm{d}P}{\mathrm{d}Q}\,|\,\mathcal{R}_t] = \mathbb{E}_Q\left[\psi_t\,|\,\mathcal{R}_t\right]$, and

$$\langle\hat{\pi}_t^{\mathcal{R}}, 1\rangle = 1 + \int_0^t \langle\hat{\pi}_s^{\mathcal{R}}, H(Y_s, \cdot, s)\rangle\mathrm{d}Y_s. \tag{30}$$

From now on, for simplicity we assume that all the processes involved in our computations are 1-dimensional. The extension to the multidimensional case is trivial. First, let us notice that, by (30) and Itô's lemma, it holds

$$\mathrm{d}\big(\langle\hat{\pi}_t^{\mathcal{R}}, 1\rangle^{-1}\big) = -\frac{\langle\hat{\pi}_t^{\mathcal{R}}, H(Y_t, \cdot, t)\rangle}{\langle\hat{\pi}_t^{\mathcal{R}}, 1\rangle^2}\mathrm{d}Y_s + \frac{\langle\hat{\pi}_t^{\mathcal{R}}, H(Y_t, \cdot, t)\rangle^2}{\langle\hat{\pi}_t^{\mathcal{R}}, 1\rangle^3}\mathrm{d}t. \tag{31}$$

Then, by the stochastic product rule,

$$\mathrm{d}\langle\pi_t^{\mathcal{R}}, \psi\rangle = \mathrm{d}\left(\langle\hat{\pi}_t^{\mathcal{R}}, \phi\rangle\langle\hat{\pi}_t^{\mathcal{R}}, 1\rangle^{-1}\right)$$

$$= \langle\hat{\pi}_t^{\mathcal{R}}, \phi\rangle\,\mathrm{d}\big(\langle\hat{\pi}_t^{\mathcal{R}}, 1\rangle^{-1}\big) + \langle\hat{\pi}_t^{\mathcal{R}}, 1\rangle^{-1}\mathrm{d}\langle\hat{\pi}_t^{\mathcal{R}}, \phi\rangle - \langle\hat{\pi}_t^{\mathcal{R}}, \phi H(Y_t, \cdot, t)\rangle\frac{\langle\hat{\pi}_t^{\mathcal{R}}, H(Y_t, \cdot, t)\rangle}{\langle\hat{\pi}_t^{\mathcal{R}}, 1\rangle^2}\mathrm{d}t$$

$$= -\langle\hat{\pi}_t^{\mathcal{R}}, \phi\rangle\frac{\langle\hat{\pi}_t^{\mathcal{R}}, H(Y_t, \cdot, t)\rangle}{\langle\hat{\pi}_t^{\mathcal{R}}, 1\rangle^2}\mathrm{d}Y_t + \langle\hat{\pi}_t^{\mathcal{R}}, \phi\rangle\frac{\langle\hat{\pi}_t^{\mathcal{R}}, H(Y_t, \cdot, t)\rangle^2}{\langle\hat{\pi}_t^{\mathcal{R}}, 1\rangle^3}\mathrm{d}t$$

$$+ \frac{\langle\hat{\pi}_t^{\mathcal{R}}, \phi H(Y_t, \cdot, t)\rangle}{\langle\hat{\pi}_t^{\mathcal{R}}, 1\rangle}\mathrm{d}Y_t - \langle\hat{\pi}_t^{\mathcal{R}}, \phi H(Y_t, \cdot, t)\rangle\frac{\langle\hat{\pi}_t^{\mathcal{R}}, H(Y_t, \cdot, t)\rangle}{\langle\hat{\pi}_t^{\mathcal{R}}, 1\rangle^2}\mathrm{d}t.$$

Recalling (26) and rearranging the terms lead us to

$$\mathrm{d}\langle\pi_t^{\mathcal{R}}, \psi\rangle = -\langle\pi_t^{\mathcal{R}}, \phi\rangle\langle\pi_t^{\mathcal{R}}, H(Y_t, \cdot, t)\rangle\mathrm{d}Y_t + \langle\pi_t^{\mathcal{R}}, \phi\rangle\langle\pi_t^{\mathcal{R}}, H(Y_t, \cdot, t)\rangle^2\mathrm{d}t$$

$$+ \langle\pi_t^{\mathcal{R}}, \phi H(Y_t, \cdot, t)\rangle\mathrm{d}Y_t - \langle\pi_t^{\mathcal{R}}, \phi H(Y_t, \cdot, t)\rangle\langle\pi_t^{\mathcal{R}}, H(Y_t, \cdot, t)\rangle\mathrm{d}t$$

$$= \left(\langle\pi_t^{\mathcal{R}}, \phi H(Y_t, \cdot, t)\rangle - \langle\pi_t^{\mathcal{R}}, \phi\rangle\langle\pi_t^{\mathcal{R}}, H(Y_t, \cdot, t)\rangle\right)\left(\mathrm{d}Y_t - \langle\pi_t^{\mathcal{R}}, H(Y_t, \cdot, t)\rangle\mathrm{d}t\right).$$

## E  PROOF OF THEOREM 5

The proof of this Theorem involves two separate parts. First, we should show the second equality in Equation (7), i.e. $\int \log \frac{\mathrm{d}P_{\#Y_{0 \le s \le t}, \phi}}{\mathrm{d}P_{\#Y_{0 \le s \le t}}\mathrm{d}P_{\#\phi}}\mathrm{d}P_{\#Y_{0 \le s \le t}, \phi} = \mathbb{E}_P\left[\log \frac{\mathrm{d}P\,|\,_{\mathcal{R}_t}}{\mathrm{d}P\,|\,_{\mathcal{F}_t^Y}\mathrm{d}P\,|\,_{\sigma(\phi)}}\right]$. Then, we should prove that the r.h.s of Equation (7) is equal to Equation (8).

### E.1  PART 1

We overload in this Section the notation adopted in the rest of the paper for sake of simplicity in exposition. A random variable $X$ on a probability space $(\Omega, \mathcal{F}, P)$ is defined as a measurable

mapping $X : \Omega \to \Psi$, where the measure space $(\Psi, \mathcal{G})$ satisfies the usual assumptions. To be precise, $X$ is measurable w.r.t. $\mathcal{F}$ if $\forall E \in \mathcal{G}, X^{-1}(E) \in \mathcal{F}$, where $X^{-1}(E) = \{\omega \in \Omega : X(\omega) \in E\}$. Equivalently, $\forall E \in \mathcal{G}, \exists S \in \mathcal{F} : X(S) = E$. Of all the possible sigma-algebras which allow measurability, the sigma algebra induced by the random variable, $\sigma(X)$, is the *smallest* one. It can be shown that $\sigma(X) = X^{-1}(\mathcal{G}) = \{A = X^{-1}(B)|B \in \mathcal{G}\}$. We also denote by $P_{\#X} : \mathcal{G} \to [0, 1]$ the push-forward measure associated to $X$ (i.e. the law), which is defined by the relation $P_{\#X}(E) = P(X^{-1}(E))$ for any $E \in \mathcal{G}$. Moreover, for any $\mathcal{G}$-measurable $\phi$, the following integration rule holds

$$\int_{\Psi} \varphi(x) dP_{\#X}(x) = \int_{\Omega} \varphi(X(\omega)) dP(\omega). \tag{32}$$

Let us focus on $(\Omega, \sigma(X), P)$ and let us consider a new measure Q absolutely continuous w.r.t. P. Radon-Nikodym theorem guarantees existence of a $\sigma(X)$-measurable function $Z : \Omega \to [0, +\infty)$ (the "derivative" $\frac{dQ}{dP} = Z$) such that $Q(A) = \int_A Z dP$, for all $A \in \sigma(X)$. Moreover, by Doob's measurability criterion (see e.g. Lemma 1.13 in Kallenberg (2002)), there exists a $\mathcal{G}$-measurable map $f : \Psi \to [0, +\infty)$ such that $Z = f(X)$. Then, for any $E \in \mathcal{G}$,

$$Q_{\#X}(E) = Q(X^{-1}(E)) = \int_{X^{-1}(E)} f(X) dP(\omega) = \int_{\Omega} \mathbf{1}_{X^{-1}(E)}(\omega) f(X(\omega)) dP(\omega)$$

$$= \int_{\Omega} \mathbf{1}_E(X(\omega)) f(X(\omega)) dP(\omega) = \int_{\Psi} \mathbf{1}_E(x) f(x) dP_{\#X}(x) = \int_E f(x) dP_{\#X}(x).$$

In summary, we have that $\frac{dQ_{\#X}}{dP_{\#X}} = f$, with $f : \Psi \to [0, +\infty)$.

Finally, then,

$$\int_{\Psi} \log\left(\frac{dP_{\#X}}{dQ_{\#X}}\right) dP_{\#X} = -\int_{\Psi} \log(f) dP_{\#X} = -\int_{\Omega} \log(f(X)) dP = \int_{\Omega} \log \frac{dP}{dQ} dP = \mathbb{E}_P[\log \frac{dP}{dQ}]. \tag{33}$$

What discussed so far, allows to prove that $\int \log \frac{dP_{\#Y_{0 \le s \le t}, \phi}}{dP_{\#Y_{0 \le s \le t}} dP_{\#\phi}} dP_{\#Y_{0 \le s \le t}, \phi} = \mathbb{E}_P\left[\log \frac{dP\,|_{\mathcal{R}_t}}{dP\,|_{\mathcal{F}_t^Y} dP\,|_{\sigma(\phi)}}\right]$. Indeed:

- Consider on the space $(\Omega, \mathcal{R}_t, P\,|_{\mathcal{R}_t})$ the random variable $T = (Y_{0 \le s \le t}, \phi)$. By construction, $\sigma(T) = \mathcal{R}_t$.
- Suppose that $P\,|_{\mathcal{R}_t}$ is absolutely continuous w.r.t $P\,|_{\mathcal{F}_t^Y} \times P\,|_{\sigma(\phi)}$ (proved in the next subsection).
- Then the desired equality follows from Equation (33).

## E.2 PART 2

Before proceeding, remember that the following holds: for all $\mathcal{R}_t' \subseteq \mathcal{R}_t$, $Q^{\mathcal{R}}\,|_{\mathcal{R}_t'} = Q^{\mathcal{R}'}\,|_{\mathcal{R}_t'}$.

We restart from the r.h.s. of Equation (7). Thanks to the chain rule for Radon-Nykodim derivatives

$$\log \frac{dP\,|_{\mathcal{R}_t}}{dP\,|_{\mathcal{F}_t^Y} dP\,|_{\sigma(\phi)}} = \log \frac{dP\,|_{\mathcal{R}_t}}{dQ^{\mathcal{R}}\,|_{\mathcal{R}_t}} \frac{dQ^{\mathcal{R}}\,|_{\mathcal{R}_t}}{dP\,|_{\mathcal{F}_t^Y} dP\,|_{\sigma(\phi)}}$$

$$= \log \frac{dP\,|_{\mathcal{R}_t}}{dQ^{\mathcal{R}}\,|_{\mathcal{R}_t}} \frac{dQ^{\mathcal{R}}\,|_{\mathcal{F}_t^Y}}{dP\,|_{\mathcal{F}_t^Y}} \frac{dQ^{\mathcal{R}}\,|_{\mathcal{R}_t}}{dQ^{\mathcal{R}}\,|_{\mathcal{F}_t^Y} dP\,|_{\sigma(\phi)}}$$

$$= \log \frac{dP\,|_{\mathcal{R}_t}}{dQ^{\mathcal{R}}\,|_{\mathcal{R}_t}} \frac{dQ^{\mathcal{F}^Y}\,|_{\mathcal{F}_t^Y}}{dP\,|_{\mathcal{F}_t^Y}} \frac{dQ^{\mathcal{R}}\,|_{\mathcal{R}_t}}{dQ^{\mathcal{R}}\,|_{\mathcal{F}_t^Y} dP\,|_{\sigma(\phi)}}$$

$$= \log \psi_t^{\mathcal{R}} (\psi_t^{\mathcal{F}^Y})^{-1} \frac{dQ^{\mathcal{R}}\,|_{\mathcal{F}_t^Y}}{dQ^{\mathcal{R}}\,|_{\mathcal{F}_t^Y} dP\,|_{\sigma(\phi)}}$$

$$= \log \psi_t^{\mathcal{R}} - \log \psi_t^{\mathcal{F}^Y} + \log \frac{dQ^{\mathcal{R}}\,|_{\mathcal{R}_t}}{dQ^{\mathcal{R}}\,|_{\mathcal{F}_t^Y} dQ^{\mathcal{R}}\,|_{\sigma(\phi)}},$$

where we used Theorem 3 to make $\psi_t^{\mathcal{R}}$ and $\psi_t^{\mathcal{F}^Y}$ appear, and the fact that $\mathrm{dQ}^{\mathcal{R}}\,|_{\sigma(\phi)} = \mathrm{dP}\,|_{\sigma(\phi)}$.

Consequently

$$
\mathbb{E}_{\mathrm{P}}\left[\log\frac{\mathrm{dP}\,|_{\mathcal{R}_t}}{\mathrm{dP}\,|_{\mathcal{F}_t^Y}\mathrm{dP}\,|_{\sigma(\phi)}}\right] = \mathbb{E}_{\mathrm{P}}\left[\log\psi_t^{\mathcal{R}} - \log\psi_t^{\mathcal{F}^Y}\right] + \mathcal{I}(Y_0;\phi)
$$

$$
= \mathbb{E}_{\mathrm{P}}\left[\int_0^t \mathbb{E}_{\mathrm{P}}[h(Y_s,X,s)\,|\,\mathcal{R}_s]\mathrm{d}W_s^{\mathcal{R}} + \frac{1}{2}\int_0^t \|\mathbb{E}_{\mathrm{P}}[h(Y_s,X,s)\,|\,\mathcal{R}_s]\|^2\mathrm{d}s\right]
$$

$$
- \mathbb{E}_{\mathrm{P}}\left[\int_0^t \mathbb{E}_{\mathrm{P}}[h(Y_s,X,s)\,|\,\mathcal{F}_s^Y]\mathrm{d}W_s^{\mathcal{F}^Y} + \frac{1}{2}\int_0^t \left\|\mathbb{E}_{\mathrm{P}}[h(Y_s,X,s)\,|\,\mathcal{F}_s^Y]\right\|^2\mathrm{d}s\right] + \mathcal{I}(Y_0;\phi)
$$

$$
= \frac{1}{2}\mathbb{E}_{\mathrm{P}}\left[\int_0^t \|\mathbb{E}_{\mathrm{P}}[h(Y_s,X,s)\,|\,\mathcal{R}_s]\|^2 - \left\|\mathbb{E}_{\mathrm{P}}[h(Y_s,X,s)\,|\,\mathcal{F}_s^Y]\right\|^2\mathrm{d}s\right] + \mathcal{I}(Y_0;\phi).
$$

Actually, the result in the main is in a slightly different form. To show equivalence, it is necessary to prove that

$$
\mathbb{E}_{\mathrm{P}}\left[\left\|\mathbb{E}_{\mathrm{P}}[h(Y_s,X,s)\,|\,\mathcal{F}_s^Y]\right\|^2\right] - 2\mathbb{E}_{\mathrm{P}}\left[\mathbb{E}_{\mathrm{P}}[h(Y_s,X,s)\,|\,\mathcal{F}_s^Y]\mathbb{E}_{\mathrm{P}}[h(Y_s,X,s)\,|\,\mathcal{R}_s]\right]
$$

$$
= -\mathbb{E}_{\mathrm{P}}\left[\left\|\mathbb{E}_{\mathrm{P}}[h(Y_s,X,s)\,|\,\mathcal{F}_s^Y]\right\|^2\right]
$$

which is trivially true since $\mathbb{E}_{\mathrm{P}}\left[\cdot\,|\,\mathcal{F}_t^Y\right] = \mathbb{E}_{\mathrm{P}}\left[\mathbb{E}_{\mathrm{P}}\left[\cdot\,|\,\mathcal{R}_s\right]\,|\,\mathcal{F}_t^Y\right]$.

# F  PROOF OF THEOREM 6

## F.1  PROOF OF EQUATION (9)

The inequality is proven considering that: i)

$$
\mathcal{I}(Y_{0\le s\le t};\phi) = \mathbb{E}_{\mathrm{P}\,|_{\mathcal{F}_t^Y}\times\mathrm{P}\,|_{\sigma(\phi)}}\left[\eta\left(\frac{\mathrm{dP}\,|_{\mathcal{R}_t}}{\mathrm{dP}\,|_{\mathcal{F}_t^Y}\mathrm{dP}\,|_{\sigma(\phi)}}\right)\right]
$$

and

$$
\mathcal{I}(\tilde{Y}_t;\phi) = \mathbb{E}_{\mathrm{P}\,|_{\sigma(\tilde{Y}_t)}\times\mathrm{P}\,|_{\sigma(\phi)}}\left[\eta\left(\frac{\mathrm{dP}\,|_{\sigma(\tilde{Y}_t,\phi)}}{\mathrm{dP}\,|_{\sigma(\tilde{Y}_t)}\mathrm{dP}\,|_{\sigma(\phi)}}\right)\right] = \mathbb{E}_{\mathrm{P}\,|_{\mathcal{F}_t^Y}\times\mathrm{P}\,|_{\sigma(\phi)}}\left[\eta\left(\frac{\mathrm{dP}\,|_{\sigma(\tilde{Y}_t,\phi)}}{\mathrm{dP}\,|_{\sigma(\tilde{Y}_t)}\mathrm{dP}\,|_{\sigma(\phi)}}\right)\right],
$$

with $\eta(x) = x\log x$, ii) that $\frac{\mathrm{dP}\,|_{\sigma(\tilde{Y}_t,\phi)}}{\mathrm{dP}\,|_{\sigma(\tilde{Y}_t)}\mathrm{dP}\,|_{\sigma(\phi)}} = \mathbb{E}_{\mathrm{P}\,|_{\mathcal{F}_t^Y}\times\mathrm{P}\,|_{\sigma(\phi)}}\left[\frac{\mathrm{dP}\,|_{\mathcal{R}_t}}{\mathrm{dP}\,|_{\mathcal{F}_t^Y}\mathrm{dP}\,|_{\sigma(\phi)}}\,|\,\sigma(\tilde{Y}_t,\phi)\right]$ and iii) that Jensen's inequality holds ($\eta$ is convex on its domain)

$$
\mathbb{E}_{\mathrm{P}\,|_{\mathcal{F}_t^Y}\times\mathrm{P}\,|_{\sigma(\phi)}}\left[\eta\left(\frac{\mathrm{dP}\,|_{\sigma(\tilde{Y}_t,\phi)}}{\mathrm{dP}\,|_{\sigma(\tilde{Y}_t)}\mathrm{dP}\,|_{\sigma(\phi)}}\right)\right]
$$

$$
= \mathbb{E}_{\mathrm{P}\,|_{\mathcal{F}_t^Y}\times\mathrm{P}\,|_{\sigma(\phi)}}\left[\eta\left(\mathbb{E}_{\mathrm{P}\,|_{\mathcal{F}_t^Y}\times\mathrm{P}\,|_{\sigma(\phi)}}\left[\frac{\mathrm{dP}\,|_{\mathcal{R}_t}}{\mathrm{dP}\,|_{\mathcal{F}_t^Y}\mathrm{dP}\,|_{\sigma(\phi)}}\,|\,\sigma(\tilde{Y}_t,\phi)\right]\right)\right]
$$

$$
\le \mathbb{E}_{\mathrm{P}\,|_{\mathcal{F}_t^Y}\times\mathrm{P}\,|_{\sigma(\phi)}}\left[\mathbb{E}_{\mathrm{P}\,|_{\mathcal{F}_t^Y}\times\mathrm{P}\,|_{\sigma(\phi)}}\left[\eta\left(\frac{\mathrm{dP}\,|_{\mathcal{R}_t}}{\mathrm{dP}\,|_{\mathcal{F}_t^Y}\mathrm{dP}\,|_{\sigma(\phi)}}\right)\,|\,\sigma(\tilde{Y}_t,\phi)\right]\right]
$$

$$
= \mathbb{E}_{\mathrm{P}\,|_{\mathcal{F}_t^Y}\times\mathrm{P}\,|_{\sigma(\phi)}}\left[\eta\left(\frac{\mathrm{dP}\,|_{\mathcal{R}_t}}{\mathrm{dP}\,|_{\mathcal{F}_t^Y}\mathrm{dP}\,|_{\sigma(\phi)}}\right)\right].
$$

## F.2  PROOF OF CONDITIONAL INDEPENDENCE AND MUTUAL INFORMATION EQUALITY

Formally the condition of conditional independence given $\pi$ is satisfied if for any $a_1, a_2$ positive random variables which are respectively $\sigma(X)$ and $\mathcal{F}_t^Y$ measurable, the following holds:

$\mathbb{E}_P[a_1 a_2 \,|\, \sigma(\pi_t)] = \mathbb{E}_P[a_1 \,|\, \sigma(\pi_t)]\mathbb{E}_P[a_2 \,|\, \sigma(\pi_t)]$ (see for instance Van Putten & van Schuppen (1985)).

The sigma-algebra $\sigma(\pi_t)$ is by definition the smallest one that makes $\pi_t$ measurable. Since $\pi_t$ is $\mathcal{F}_t^Y$ measurable, clearly $\sigma(\pi_t) \subseteq \mathcal{F}_t^Y$. By the very definition of conditional expectation, $\mathbb{E}_P[a_1 \,|\, \mathcal{F}_t^Y] = \langle \pi_t, a_1 \rangle$, which is an $\sigma(\pi_t)$ measurable quantity. Then $\mathbb{E}_P[a_1 a_2 \,|\, \sigma(\pi_t)] = \mathbb{E}_P[\mathbb{E}_P[a_1 a_2 \,|\, \mathcal{F}_t^Y] \,|\, \sigma(\pi_t)] = \mathbb{E}_P[\mathbb{E}_P[a_1 \,|\, \mathcal{F}_t^Y]a_2 \,|\, \sigma(\pi_t)] = \mathbb{E}_P[\mathbb{E}_P[\langle \pi_t, a_1 \rangle a_2 \,|\, \sigma(\pi_t)] = \langle \pi_t, a_1 \rangle \mathbb{E}_P[a_2 \,|\, \sigma(\pi_t)]$. Since $\langle \pi_t, a_1 \rangle = \mathbb{E}_P[\langle \pi_t, a_1 \rangle \,|\, \sigma(\pi_t)] = \mathbb{E}_P[\mathbb{E}_P[a_1 \,|\, \mathcal{F}_t^Y] \,|\, \sigma(\pi_t)] = \mathbb{E}_P[a_1 \,|\, \sigma(\pi_t)]$, the proof of conditional independence is concluded.

In summary, $\sigma(X)$ and $\mathcal{F}_t^Y$ are conditionally independent given $\sigma(\pi_t)$ ($\subset \mathcal{F}_t^Y$). This implies that $P(A \,|\, \sigma(\pi_t)) = P(A \,|\, \mathcal{F}_t^Y)$, $\forall A \in \sigma(X)$, or equivalently $\mathbb{E}_P[\mathbf{1}(A) \,|\, \sigma(\pi_t)] = \mathbb{E}_P[\mathbf{1}(A) \,|\, \mathcal{F}_t^Y]$. To prove this, it is sufficient to show that for any $B \in \mathcal{F}_t^Y$, $\mathbb{E}_P[\mathbb{E}_P[\mathbf{1}(A) \,|\, \sigma(\pi_t)]\mathbf{1}(B)] = \mathbb{E}_P[\mathbf{1}(A)\mathbf{1}(B)]$. By standard properties of conditional expectation $\mathbb{E}_P[\mathbb{E}_P[\mathbf{1}(A) \,|\, \sigma(\pi_t)]\mathbf{1}(B)] = \mathbb{E}_P[\mathbb{E}_P[\mathbf{1}(A) \,|\, \sigma(\pi_t)]\mathbb{E}_P[\mathbf{1}(B) \,|\, \sigma(\pi_t)]]$. Due to conditional independence $\mathbb{E}_P[\mathbf{1}(A) \,|\, \sigma(\pi_t)]\mathbb{E}_P[\mathbf{1}(B) \,|\, \sigma(\pi_t)] = \mathbb{E}_P[\mathbf{1}(A)\mathbf{1}(B) \,|\, \sigma(\pi_t)]$. Then, $\mathbb{E}_P[\mathbb{E}_P[\mathbf{1}(A) \,|\, \sigma(\pi_t)]\mathbb{E}_P[\mathbf{1}(B) \,|\, \sigma(\pi_t)]] = \mathbb{E}_P[\mathbb{E}_P[\mathbf{1}(A)\mathbf{1}(B) \,|\, \sigma(\pi_t)]] = \mathbb{E}_P[\mathbf{1}(A)\mathbf{1}(B)]$.

The mutual information equality is then proved considering that $\frac{dP \,|\, \mathcal{R}_t}{dP \,|\, \mathcal{F}_t^Y \, dP \,|\, \sigma(\phi)} = \frac{dP(\omega^x \,|\, \mathcal{F}_t^Y)}{dP(\omega^x)}$, since the conditional probabilities exist, and that $P(\omega^x \,|\, \mathcal{F}_t^Y) = P(\omega^x \,|\, \sigma(\pi_t))$.

## G   A TECHNICAL NOTE

As anticipated in the main, Assumption 1 might be incompatible with the other technical assumptions in Appendix A. The problem might arise for singularities in the drift term at time $t = T$, which are usually present in the construction of dynamics satisfying Assumption 1 like stochastic bridges. This mathematical subtlety can be more clearly interpreted by noticing that when Assumption 1 is satisfied the evolution of the posterior process $\pi_t$ at time $T$ can occupy a portion of the space of dimensionality lower than at any $T - \epsilon, \epsilon > 0$. Or, we can notice that if Assumption 1 is satisfied, $\mathcal{I}(Y_{0 \leq s \leq T}; V) = \mathcal{I}(V; V)$ which can be equal to infinity depending on the actual structure of $\mathcal{S}$ and the mapping $V$. In many cases, a simple technical solution is to consider in the analysis only dynamics of the process in the time interval $[0, T)^3$. In the reduced time interval $[0, T)$, the technical assumptions are generally shown to be satisfied. For the practical purposes explored in this work this restriction makes no difference, and consequently neglect it for the rest of our discussion.

## H   LINEAR DIFFUSION MODELS

Consider the particular case of **linear** generative diffusion models Song et al. (2021), which are widely adopted in the literature and by practitioners. We consider the particular case of Equation (11), where the function $F$ has linear expression

$$\hat{Y}_t = \hat{Y}_0 - \alpha \int_0^t \hat{Y}_s ds + \hat{W}_t, \tag{34}$$

for a given $\alpha \geq 0$. We assume of course again that Assumption 1 holds, which implies that we should select $\hat{Y}_0 = Y_T = V$. Now, $\alpha$ dictates the behavior of the SDE, which can be cast to the so called Variance-Preserving and Variance Exploding schedules of diffusion models Song et al. (2021). In diffusion models jargon, Equation (34) is typically referred to as a *noising* process. Indeed, by analysing the evolution of Equation (34), $\hat{Y}_t$ evolves to a noisier and noisier version of $V$ as $t$ grows. In particular, it holds that

$$\hat{Y}_t = \exp(-\alpha t)V + \exp(-\alpha t)\int_0^t \exp(\alpha s)d\hat{W}_s.$$

---

[3]This is akin to the discussion of *arbitrage* strategies in finance when the initial filtration is augmented with knowledge of the future value at certain time instants, and the fact that while the new process adapted w.r.t the new filtration is also a martingale w.r.t. a given new measure for all $t \in [0, T)$, it fails to do so for $t = T$ (thus giving an arbitrage opportunity).

The next result is a particular case of Theorem 7.

**Lemma 1.** *Consider the stochastic process $Y_t$ which solves Equation (34). The same stochastic process also admits a $\mathcal{F}_t^Y$ –adapted representation*

$$Y_t = Y_0 + \int_0^t \alpha Y_s + 2\alpha \frac{\exp(-\alpha(T-s))\mathbb{E}_{\mathrm{P}}[V \mid \sigma(Y_s)] - Y_s}{1 - \exp(-2\alpha(T-s))} \mathrm{d}s + W_t, \tag{35}$$

*where $Y_0 = \exp(-\alpha T)V + \sqrt{\frac{1-\exp(-2\alpha T)}{2\alpha}}\epsilon$, with $\epsilon$ a standard Gaussian random variable independent of $V$ and $W_t$.*

As discussed in the main paper, we can now show that the same generative dynamics can be obtained under the NLF framework we present in this work, without the need to explicitly defining a backward and a forward process. In particular, we can directly select a observation function that corresponds to an Orstein-Uhlenbeck bridge (Mazzolo, 2017; Corlay, 2013), consequently satisfying Assumption 1, and obtain the generative dynamics of classical diffusion models. In particular we consider the following about $H^4$:

**Assumption 7.** *The function $H$ in Equation (1) is selected to be of the linear form*

$$H(Y_t, X, t) = m_t V - \frac{\mathrm{d}\log m_t}{\mathrm{d}t} Y_t, \tag{36}$$

*with $m_t = \frac{\alpha}{\sinh(\alpha(T-t))}$, where $\alpha \geq 0$. When $\alpha = 0$, $m_t = \frac{\mathrm{d}\log m_t}{\mathrm{d}t} = \frac{1}{T-t}$. Furthermore, $Y_0$ is selected as in Theorem 7. Under this assumption, $Y_T = V$, $\mathrm{P} - a.s.$, i.e. Assumption 1 is satisfied [Proof].*

In summary, the particular case of Equation (1) (which is $\mathcal{F}^{Y,X}$ adapted) under Assumption 7, can be transformed into a generative model leveraging Theorem 2, since Assumption 1 holds. When doing so, we obtain that the process $Y_t$ has $\mathcal{F}^Y$ adapted representation equal to

$$Y_t = Y_0 + \int_0^t m_s \mathbb{E}_{\mathrm{P}}(V \mid \mathcal{F}_s^Y)\mathrm{d}s - \int_0^t \frac{\mathrm{d}\log m_s}{\mathrm{d}s} Y_s \mathrm{d}s + W_t^{\mathcal{F}^Y}, \tag{37}$$

which is nothing but Equation (35) after some simple algebraic manipulation. The only relevant detail worth deeper exposition is the clarification about the actual computation of expectation of interest. If P is selected such that $\hat{Y}_t$ solves Equation (34), we have that

$$\mathbb{E}_{\mathrm{P}}(V \mid \mathcal{F}_t^Y) = \mathbb{E}_{\mathrm{P}}(Y_T \mid \sigma(Y_{0 \leq s \leq t})) = \mathbb{E}_{\mathrm{P}}(\hat{Y}_0 \mid \sigma(\hat{Y}_{T-t \leq s \leq T})) = \mathbb{E}_{\mathrm{P}}(\hat{Y}_0 \mid \sigma(\hat{Y}_{T-t})) = \mathbb{E}_{\mathrm{P}}(V \mid \sigma(Y_t)), \tag{38}$$

where the second to last equality is due to the Markov nature of $\hat{Y}_t$.

Moreover, in this particular case we can express the mutual information $\mathcal{I}(Y_{0 \leq s < t}; \phi) = \mathcal{I}(Y_t; \phi)$ ( where we removed the past of $Y$ since the following Markov chain holds $\phi \to \hat{Y}_0 \to \hat{Y}_{t>0}$) can be expressed in the simpler form

$$\mathcal{I}(Y_t; \phi) = \mathcal{I}(Y_0; \phi) + \frac{1}{2}\mathbb{E}_{\mathrm{P}}\left[\int_0^t m_s^2 \|\mathbb{E}_{\mathrm{P}}[V \mid \sigma(Y_s)] - \mathbb{E}_{\mathrm{P}}[V \mid \sigma(Y_s, \phi)]\|^2 \mathrm{d}s\right] \tag{39}$$

matching the result described in Franzese et al. (2023), obtained with the formalism of time reversal of SDEs.

# I  DISCUSSION ABOUT ASSUMPTION 7

This is easily checked thanks to the following equality

$$Y_t = Y_0 \frac{m_0}{m_t} + V \frac{m_0}{m_{T-t}} + \int_0^t \frac{m_s}{m_t} \mathrm{d}W_s. \tag{40}$$

---

[4]Notice that with $H$ selected as in Assumption 7 the validity of the theory considered is restricted to the time interval $[0, T)$, see also Appendix G.

To avoid cluttering the notation, we define $f_t = \frac{\mathrm{d}\log m_t}{\mathrm{d}t}$. To show that Equation (40) is true, it is sufficient to observe i) that initial conditions are met and ii) that the time differential of the process is the correct one. We proceed to show that indeed the second condition holds (the first one is trivially observed to be true).

$$
\begin{aligned}
\mathrm{d}Y_t &= -\alpha Y_0 \frac{\cosh(\alpha(T-t))}{\sinh(\alpha T)} + \alpha r(X)\frac{\cosh(\alpha t)}{\sinh(\alpha T)} - \alpha\cosh(\alpha(T-t))\int_0^t \frac{1}{\sinh(\alpha(T-s))}\mathrm{d}W_s + \mathrm{d}W_t \\
&= -\alpha\frac{\cosh(\alpha(T-t))}{\sinh(\alpha(T-t))}\left(Y_0\frac{\sinh(\alpha(T-t))}{\sinh(\alpha T)} + \int_0^t \frac{\sinh(\alpha(T-t))}{\sinh(\alpha(T-s))}\mathrm{d}W_s\right) + \alpha r(X)\frac{\cosh(\alpha t)}{\sinh(\alpha T)} + \mathrm{d}W_t \\
&= -\alpha\coth(\alpha(T-t))\left(Y_t - r(X)\frac{\sinh(\alpha t)}{\sinh(\alpha T)}\right) + \alpha r(X)\frac{\cosh(\alpha t)}{\sinh(\alpha T)} + \mathrm{d}W_t \\
&= -f_t Y_t + \alpha r(X)\left(\frac{\coth(\alpha(T-t))\sinh(\alpha t)}{\sinh(\alpha T)} + \frac{\cosh(\alpha t)}{\sinh(\alpha T)}\right) + \mathrm{d}W_t \\
&= -f_t Y_t + \alpha r(X)\left(\frac{\coth(\alpha(T-t))\sinh(\alpha t) + \cosh(\alpha t)}{\sinh(\alpha T)}\right) + \mathrm{d}W_t \\
&= -f_t Y_t + \alpha r(X)\left(\frac{\coth(\alpha(T-t))\sinh(\alpha t) + \cosh(\alpha t)}{\sinh(\alpha T)}\right) + \mathrm{d}W_t \\
&= -f_t Y_t + m_t r(X) + \mathrm{d}W_t
\end{aligned}
$$

where the result is obtained considering that

$$
\begin{aligned}
\frac{\coth(\alpha(T-t))\sinh(\alpha t) + \cosh(\alpha t)}{\sinh(\alpha T)} &= \frac{\frac{e^{\alpha(T-t)}+e^{-\alpha(T-t)}}{e^{\alpha(T-t)}-e^{-\alpha(T-t)}}\left(e^{\alpha t}-e^{-\alpha t}\right) + \left(e^{\alpha t}+e^{-\alpha t}\right)}{e^{\alpha T}-e^{-\alpha T}} \\
&= \frac{\frac{e^{\alpha T}+e^{-\alpha(T-2t)}-e^{\alpha(T-2t)}-e^{-\alpha T}}{e^{\alpha(T-t)}-e^{-\alpha(T-t)}} + \left(e^{\alpha t}+e^{-\alpha t}\right)}{e^{\alpha T}-e^{-\alpha T}} \\
&= \frac{e^{\alpha T}+e^{-\alpha(T-2t)}-e^{\alpha(T-2t)}-e^{-\alpha T}+e^{\alpha T}-e^{-\alpha(T-2t)}+e^{\alpha(T-2t)}-e^{-\alpha T}}{\left(e^{\alpha(T-t)}-e^{-\alpha(T-t)}\right)\left(e^{\alpha T}-e^{-\alpha T}\right)} \\
&= \frac{2}{e^{\alpha(T-t)}-e^{-\alpha(T-t)}}.
\end{aligned}
$$

## J  EXPERIMENTAL DETAILS

### J.1  DATASET DETAILS

The Shapes3D dataset (Kim & Mnih, 2018) includes the following attributes and the number of classes for each, as shown in Table 1.

**Table 1:** Attributes and class counts in the Shapes3D dataset.

| Attribute | Number of Classes |
|---|---|
| Floor hue | 10 |
| Object hue | 10 |
| Orientation | 15 |
| Scale | 8 |
| Shape | 4 |
| Wall hue | 10 |

### J.2  UNCONDITIONAL DIFFUSION MODEL TRAINING

We train the unconditional denoising score network using the NCSN++ architecture (Song et al., 2021), which corresponds to a U-NET (Ronneberger et al., 2015). The model is trained from scratch using the score-matching objective. The training hyperparameters are summarized in Table 2.

Table 2: Hyperparameters for unconditional diffusion model training.

| Parameter | Value |
|---|---|
| Epochs | 100 |
| Batch size | 256 |
| Learning rate | $1 \times 10^{-4}$ |
| Optimizer | AdamW (Loshchilov & Hutter, 2019) |
| $\beta_1$ | 0.95 |
| $\beta_2$ | 0.999 |
| Weight decay | $1 \times 10^{-6}$ |
| Epsilon | $1 \times 10^{-8}$ |
| Learning rate scheduler | Cosine annealing with warmup |
| Warmup steps | 500 |
| Gradient clipping | 1.0 |
| EMA decay | 0.9999 |
| Mixed precision | FP16 |
| Scheduler | Variance Exploding (Song et al., 2021) |
| $\sigma_{\min}$ | 0.01 |
| $\sigma_{\max}$ | 90 |
| Loss function | Denoising score matching (Song et al., 2021) |

## J.3 LINEAR PROBING EXPERIMENT DETAILS

In the linear probing experiments, we train a linear classifier on the feature maps extracted from the denoising score network at various noise levels $\tau$. The training details are provided in Table 3.

Table 3: Hyperparameters for linear probing experiments.

| Parameter | Value |
|---|---|
| Batch size | 64 |
| Loss function | Cross-Entropy Loss |
| Optimizer | Adam (Kingma & Ba, 2015) |
| Learning rate | $1 \times 10^{-6}$ for $\tau = 0.9$ or $\tau = 0.99$ 
 $1 \times 10^{-4}$ for other $\tau$ values |
| Number of epochs | 30 |
| Inputs | Feature maps (used as-is in the linear layer) 
 Noisy images (scaled to $[-1, +1]$) |

## J.4 MUTUAL INFORMATION ESTIMATION EXPERIMENT DETAILS

For mutual information estimation, we train a conditional diffusion model using the same NCSN++ architecture as before. The conditioning is incorporated by adding a distinct class embedding for each label present in the input image, added to the input embedding along with the timestep embedding. The hyperparameters are the same as those used for the unconditional diffusion model (see Table 2).

To calculate the mutual information, we use Equation 39, estimating the integral using the midpoint rule with 999 points uniformly spaced in $[0, T]$.

## J.5 FORKING EXPERIMENT DETAILS

In the forking experiments, we use a ResNet50 (He et al., 2016) model with an additional linear layer, trained from scratch, to classify the generated images and assess label coherence across forks. The training details for the classifier are summarized in Table 4.

During the sampling process of the forking experiment, we use the settings summarized in Table 5.

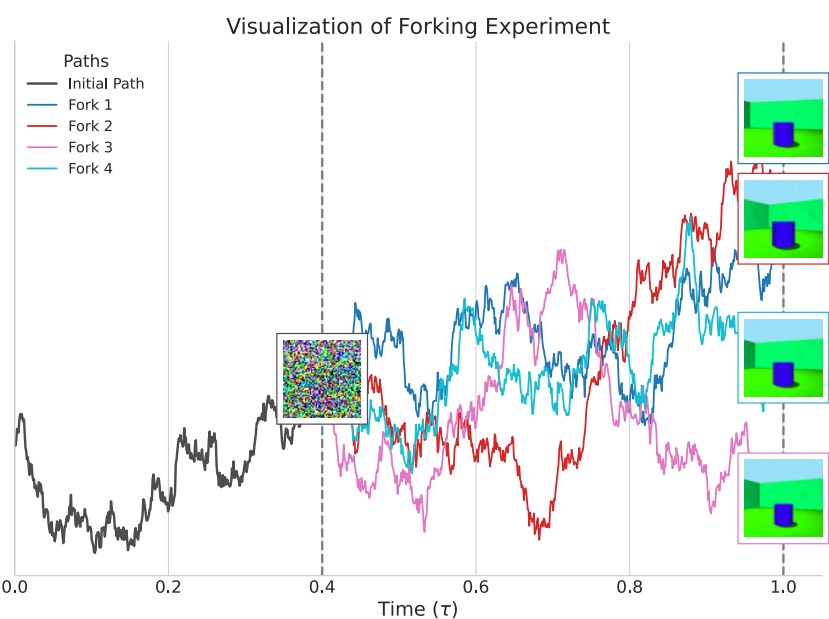

**Figure 4:** Visualization of the forking experiment with `num_forks = 4` and one initial seed. The image at time $\tau = 0.4$ is quite noisy. In the final generations after forking, the images exhibit coherence in the labels *shape*, *wall hue*, *floor hue*, and *object hue*. However, there is variation in *orientation* and *scale*.

**Table 4:** Hyperparameters for the classifier in forking experiments.

| Parameter | Value |
|---|---|
| Image size | 224 (resized with bilinear interpolation) |
| Image scaling | $[-1, +1]$ |
| Dataset split | Training set: 72% |
| | Validation set: 8% |
| | Test set: 20% |
| Early stopping | Stop when validation accuracy exceeds 99% |
| | Evaluated every 1000 steps |
| Number of epochs | 1 |
| Optimizer | Adam (Kingma & Ba, 2015) |
| Learning rate | $1 \times 10^{-4}$ |

**Table 5:** Sampling settings for the forking experiments.

| Parameter | Value |
|---|---|
| Stochastic predictor | Euler-Maruyama method with 1000 steps |
| Corrector | Langevin dynamics with 1 step |
| Signal-to-noise ratio (SNR) | 0.06 |
| Number of forks ($k$) | 100 |
| Number of seeds | 10 (independent initial noise samples) |

## J.6 LINEAR PROBING ON RAW DATA

In Figure 5, we evaluate the performance of linear probes trained on features maps extracted from the denoiser network, and show compare their log probability accuracy with a linear probe trained on the raw, noisy input and a random guesser. Throughout the generative process, linear probes obtain higher accuracy than the baselines: for large noise levels, a linear probe on raw input data fails, whereas the inner layers of the denoising network extract features that are sufficient to discern latent labels.

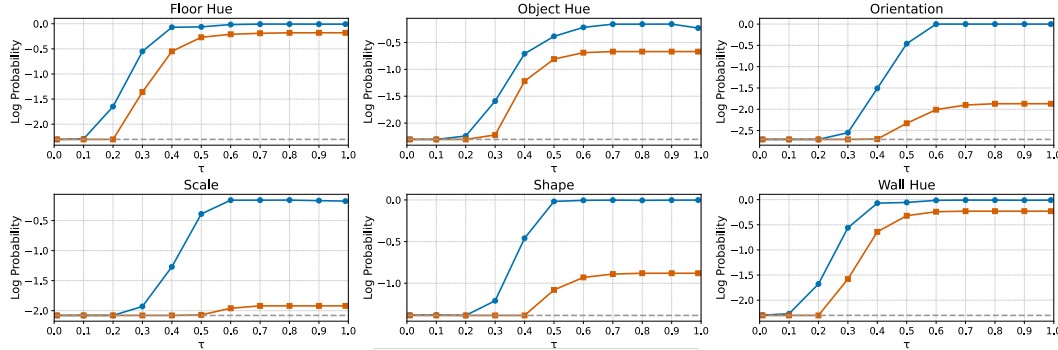

**Figure 5:** Log-probability accuracy of linear classifiers at $\tau$. 'Feature map' classifiers are trained on network features; 'Noisy Image' trained on noisy images; 'Random Guess' is the baseline for random guessing.

## J.7 ADDITIONAL EXPERIMENTS ON CELEBA DATASET

We present our results conducted on the CelebA dataset (Liu et al., 2015), consisting of over 200000 celebrity images with 40 binary attributes. Next, we focus our analysis on the attributes "Male" and "Eyeglasses" as these are i) among the most reliable and objectively labeled features in the CelebA dataset[5] and ii) significant examples of attributes which can be mapped to more global and local features respectively. The unconditional and conditional diffusion models were trained using the identical architectural, optimization, and training hyperparameters as in Song et al. (2021). Both models employed a variance-exploding diffusion process with a U-Net backbone for the denoising score network. Training details, including the learning rate, batch size, and noise schedules, are the same as of Song et al. (2021). We present a comprehensive analysis of the results derived from probing experiments, mutual information (MI) estimation, and the rate of increase of MI across the generative process.

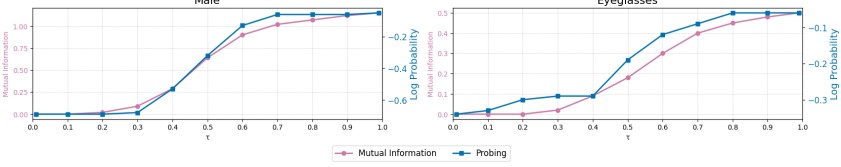

**Figure 6:** Probing accuracy and mutual information (MI) as a function of the noise intensity parameter $\tau$.

**Probing vs. MI.** Our results, as shown in Figure 6, illustrate a coherent growth between classifier accuracy (probing performance) and mutual information as a function of the noise intensity parameter $\tau$. For both attributes, probing accuracy increases steadily, mirroring the growth of MI.

**Mutual Information Across Labels** Figure 7 compares MI growth across the "Male" and "Eyeglasses" attributes. A key observation is that the MI for "Male" rises earlier than for "Eyeglasses", beginning at $\tau = 0.2$, compared to $\tau = 0.3$. This aligns with the intuition that some latent abstractions emerge earlier in the generative process than others, given that the average number of pixels impacted by the global features is larger than the local ones.

---

[5]This is supported by previous work, which highlights significant labeling issues for many other attributes, making them less suitable for consistent analysis (Lingenfelter et al., 2022).

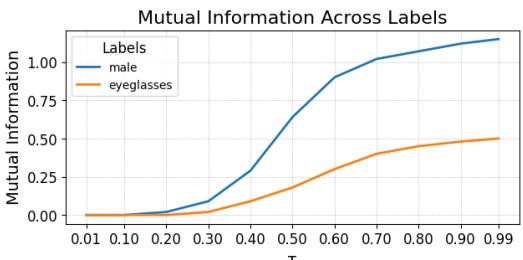

**Figure 7:** Mutual information (MI) growth for "Male" and "Eyeglasses" attributes across the generative process.

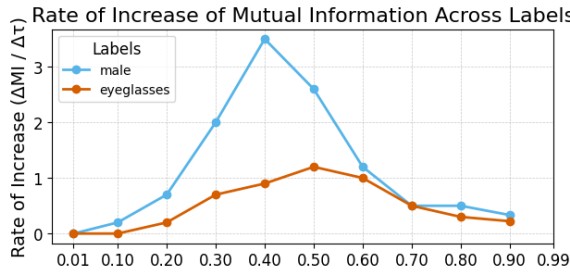

**Figure 8:** Rate of change of mutual information (MI) for "Male" and "Eyeglasses" attributes as a function of $\tau$.

**Rate of Increase of MI** To further investigate the dynamics, we plot $\frac{\Delta(MI)}{\Delta\tau}$, the rate of change of MI, for the two attributes (Figure 8). This reveals that "Male" exhibits a significantly faster initial growth rate compared to "Eyeglasses", peaking around $\tau = 0.4$. This confirms the earlier emergence of "Male" as a latent abstraction, with a sharp rise in MI during the early stages. In contrast, the MI for "Eyeglasses" grows more gradually, reflecting a slower but steady emergence of this attribute.

