# OpenReview forum: "Latent Abstractions in Generative Diffusion Models"
_ICLR.cc/2025/Conference — Submitted to ICLR 2025_

### Official Review · Reviewer_w2Gh · 2024-11-01

**Soundness:** 3
**Presentation:** 2
**Contribution:** 3
**Rating:** 6
**Confidence:** 3

**Summary:**

This work studies the role of latent abstractions in the stochastic general process of diffusion models by extending the nonlinear filtering (NLF) framework. More specifically, this work proves that the generative process can be viewed as a switch process between $Y_t$ and $\pi_t$, which corresponds to the image and latent abstractions space. Furthermore, the synthetic and real-world experiments support their theoretical results.

**Strengths:**

1.	The link between the NLF and diffusion models is really interesting.
2.	This work includes theoretical and empirical parts, which help us understand the role of latent abstractions.

**Weaknesses:**

1.	There are many theorems in the introduction part, which makes it hard to find the most important results of this work. It would be better to polish the presentation of this work. More specifically, I think it would be helpful to present Eq. (10) and Figure 1 and introduce the relationship between Eq. (10) and the classic stochastic process.
2.	It would be better to discuss the relationship between the experiment part and the theoretical part. It seems that the only connection is the mutual information estimation part.

**Questions:**

Please see the Weakness part.

Q1: Could this NLF framework be extended to the deterministic sampling process?

---

> ### Author Response · Authors · 2024-11-21
> **Rebuttal**
>
> We thank the reviewer for the thoughtful feedback and for recognizing the strengths of our work, particularly the connection between nonlinear filtering (NLF) and diffusion models and the complementarity of theoretical and empirical components. Below, we address the reviewer's observations in detail.
>
> We understand the reviewer’s concern regarding the presentation of key results and appreciate the suggestion to prioritize Eq. (10) and Figure 1. However, we found during the drafting process of our paper that presenting these elements earlier, without proper introduction, would leave the reader without the necessary context to understand their significance. The sequence of presenting the fundamentals first, followed by their extension to the operational mutual information (MI) equation, is essential to maintain clarity and rigor in our exposition.
>
> The connection between theory and experiments indeed hinges on mutual information, as the reviewer interestingly points out. This connection is at the heart of our contribution. While our experiments focus on diffusion models (to reach a broader audience), the theoretical results are intentionally presented in their full generality to emphasize their applicability to a broader class of generative models. The rigorous presentation of the fundamentals ensures that our results are not limited to the specific case of diffusion models but are valid for any model satisfying Assumption 1. **Ultimately, we believe that the merit of our work is that of providing solid theoretical foundations for further research and applications on a broad class of generative models.**
>
> *Q1: Could this NLF framework be extended to the deterministic sampling process?*
>
> Thank you for the interesting question! It is indeed possible to extend the NLF framework to deterministic sampling processes. However, such an extension requires careful consideration of the differences between deterministic and stochastic dynamics. Specifically:
> - In deterministic processes, the mutual information between $Y_t$ and $X$ remains constant through time and is equal to its maximum from the outset. This is because the value of the dynamics of the process $Y$ at a given time $t'$ can be obtained from any other value at a different time $t$ (the mapping $Y_t \rightarrow Y_{t'}$ is deterministic and known)
> - Despite this, our formalism remains relevant, as it helps to analyze when and how such information becomes linearly accessible in the activation space of the architecture under consideration. This is crucial for understanding the role of latent abstractions, even in deterministic settings.
>
> This extension highlights the versatility of our framework and its applicability to a wide range of generative models, including those utilizing deterministic sampling schemes.

---

> > ### Comment · Reviewer_w2Gh · 2024-11-26
> >
> > Thanks for your response and real-world experiments. It is a very nice thing to provide a unified theoretical framework that goes beyond diffusion models (Schrödinger Bridges, and stochastic normalizing flows). However, I still think it is helpful to present the paper in a way that is friendly to reader. I will matains my score.

---

### Official Review · Reviewer_UtpZ · 2024-11-03

**Soundness:** 3
**Presentation:** 3
**Contribution:** 3
**Rating:** 6
**Confidence:** 3

**Summary:**

This work offers a new way to theoretically understand the generative process of diffusion models using nonlinear filtering. In particular, the authors interpret the generative process in diffusion model as simulating the NLF dynamics, where an unobservable latent abstraction drives observable outcomes in the generative process. Second, the authors propose a concrete method to compute the mutual information between measurement processes and latent abstractions, measuring the extent to which a certain latent variable has been constructed at different stages of the generative process. Third, experiments have been conducted using the standard denoising diffusion model on the Shapes3D data with 6 attributes (floor hue, wall hue, object hue, scale, shape, orientations) as labels. Apart from mutual information, the authors also use the log likelihood (using a trained classifier network) and the entropy in a forking experiments, as proxies for latent variable abstraction.

**Strengths:**

The NLF viewpoint on diffusion model is to my knowledge new, and potentially interesting. The use of mutual information measurement is also helpful in quantifying the effect of noising/denoising on the latent variables (here labels). The experiments seem carefully carried out.

**Weaknesses:**

The methodology is sound provided that the datasets can be understood as being dictated by a low-dimensional set of latent variables, and that it is possible to understand the generative process as inferring the latent variables in a Baysian way. The experiments are also conducted solely on such a synthetic dataset, with each sample having a predetermined set of labels. Either the including of a theoretical discussion on this underlying basic assumption, or devising experiments on more diverse datasets, preferably both, will make the relevance of the work more convincing.

**Questions:**

1. The work by Scolocchi et al that is cited in the manuscript seems to contain many similar claims and the forward-backward experiments there seem similar to the forking experiments here, though the theoretical methods are different. Can the authors comment more on the differences and similarities and possible relations between the two works in the paper?
2.  The caption of Figure 2 (noise with intensity $\tau$) seems to contradict ths figure?
3. In line 405, do you mean "$\pi_3$ contains the same information... as the whole ...."?
4. The results such as those shown in Figure 3 are given as a function of $\tau$, but I assume the meaning of which depends on the noising schedule chosen for the diffusion model. Wouldn't it give more insight to discuss them with respect to intrinsic quantities such as the signal-to-noise ration instead of the "time" $\tau$?

---

> ### Author Response · Authors · 2024-11-21
> **Rebuttal**
>
> We thank the reviewer for the thoughtful feedback and for highlighting important points for discussion. Below, we address them in detail.
>
> **Relationship Between Measurements and Latent Abstractions**
>
> We agree with the reviewer on the importance of clarifying the relationship between measurements and latent abstractions. This is discussed in lines 323–324, where we explain how to transition from $\mathcal{F}^Y$ (filtration based on measurements) to an enlarged filtration. To summarize, there are two possible cases:
>
> - Known Latent World “Ingredients”: If the latent world is known, we have a generative model with an equation in the form of (1), which directly incorporates latent abstractions.
> - Unknown Latent World “Ingredients”: If the generative process works purely on measurements, latent abstractions can still be introduced mathematically by enlarging the filtration. This step allows us to reason about the hierarchical emergence of abstractions while keeping the theoretical framework general.
>
> We will make this connection more explicit in the revised version of the paper to address this point thoroughly, thank you for the suggestion!
>
> To further corroborate our experimental findings, we are currently running experiments on the CelebA dataset, where we are observing a similar pattern of hierarchical feature emergence. For example, high-level attributes, such as gender, emerge earlier in the generative process than finer-grained details, such as the presence of eyeglasses, which only appear at later stages. We will place our additional results in the appendix of the camera ready version of the paper.
>
> **Relation to Sclocchi et al. (2024)**
>
> We acknowledge the connection between our work and the study by Sclocchi et al. (2024), and we appreciate the opportunity to further clarify the differences and similarities.
>
> Key distinctions are as follows:
> - **Scope of Analysis**: The work by Sclocchi et al. focuses specifically on generative diffusion models, while our theoretical framework is broader. As discussed in our response to Reviewer PspT, any model satisfying Assumption 1 can be analyzed using our information-theoretic framework. Diffusion models are a significant instance, but not the sole focus of our analysis.
> - **Hierarchical Structure**: Sclocchi et al. assume an explicit hierarchical structure for the data (see their Figure 3) and use mean-field theory to study the inference of such variables during the generative process. In contrast, our framework does not assume a pre-defined hierarchy of latent variables. Instead, we show that a hierarchy naturally arises during the generative process, as explained by Eq. (8) and the discussion in lines 224–232. This emergence is a direct consequence of the information-theoretic dynamics of the generative process and does not rely on a specific structural assumption.
> Note also that our work provides experimental evidence of this phenomenon, see the results obtained in our Figure 3 on the Shapes3D dataset. The attributes that occupy the largest portion of the space (wall hue and floor hue) behave similarly and are the first to appear in the generation process, at $\tau=0.4$. The object hue emerges after these two, at $\tau=0.5$ but before orientation, scale and shape, which appear at $\tau=0.6$. Intuitively, given the nature of the dataset, this hierarchy is reasonable, since (for example) varying the orientation of a shape once the color is fixed, has a lower impact in terms of norms, than changing the color of an object.
>
> - **Experimental Comparison**: While the forward-backward experiments in Sclocchi et al. may seem similar to our forking experiments, they are conceptually different. In particular, in the experimental settings described by their Figure 2, it is possible to observe how via a procedure similar to our forking experiments the cosine similarity of the activation functions in a reference ConvNeXt Base vary as a function of the diffusion time $t$.
> Our forking experiments directly probe when and how latent factors emerge by analyzing coherence across generative pathways and compare such coherence with i) the accuracy of a linear classifier on the hidden activations of the score network and ii) with the mutual information.
>
>
> While the two works are clearly connected, they attack different problems with different techniques. We will revise the manuscript to better highlight these points and further clarify the relationship between the two works.

---

> > ### Author Response · Authors · 2024-11-21
> > **.**
> >
> > **Caption of Figure 2 and Noise Intensity**
> > We agree with the reviewer that the caption of Figure 2 can be improved. The intensity of the noise is indeed zero at the end of the generative process, and we appreciate this observation. We will amend the figure caption to avoid confusion.
> >
> > We appreciate the reviewer’s suggestion regarding the use of intrinsic quantities such as signal-to-noise ratio (SNR), instead of time $t$, in results like those shown in Figure 3. However, we believe that retaining the reference to a generic time variable is important for several reasons:
> > - **Generality Beyond Diffusion Models**: While SNR is meaningful in the context of diffusion models, it is not straightforward to define or interpret it for other types of generative models (e.g., stochastic normalizing flows, neural SDEs, or Schrödinger Bridges).
> > - **Consistency with Theoretical Framework**: Our framework generalizes to models where $t$ plays a role in the evolution of dynamics without necessarily tying it to noise levels. Using time as the reference variable maintains this generality.
> >
> > To improve clarity, we will  revise the paper to explicitly discuss the relationship between time and SNR, while emphasizing the general applicability of our framework.
> >
> >
> > **Typographical Error in Line 405**
> > Thank you for spotting the typo. Your interpretation is indeed correct, and we will fix it in our revision.
> >
> > We appreciated the opportunity to address these points and thank the reviewer again for the careful review.

---

### Official Review · Reviewer_VD8o · 2024-11-03

**Soundness:** 3
**Presentation:** 3
**Contribution:** 3
**Rating:** 6
**Confidence:** 2

**Summary:**

This paper introduces a theoretical framework that extends Nonlinear Filtering (NLF) to explain the emergence and influence of latent abstractions in diffusion-based generative models. Through the lens of stochastic differential equations (SDEs), the authors model the dynamics of observable and latent variables, showing how latent abstractions emerge hierarchically during the generative process. The study derives an information-theoretic measure to quantify the influence of these latent factors and validates the theory with experiments demonstrating distinct jumps in this measure, supporting the proposed model.

**Strengths:**

* Well-motivated introduction, situating the work within existing literature and highlighting its contributions.
* The structure is logically organized, and the mathematical formulations are rigorous, enhancing the soundness of the theory.
* The proposed information-theoretic measure effectively captures changes in latent abstractions during generation, with experiments showing jumps in the measure that align with theoretical expectations.

**Weaknesses:**

I have no significant weaknesses to report. Please see questions for minor points of clarification.

**Questions:**

* Lines 228-233: could you expand on the claim regarding hierarchical emergence of latent factors? For instance, would an experiment focusing on local details, such as grain on texture, further validate this? If the mutual information of these details increased later in the process, it would support this hierarchical perspective.
* For the mutual information estimation process on line 476, could you clarify the final numerical approach used to approximate this measure during the generative process?

---

> ### Author Response · Authors · 2024-11-21
> **Rebuttal**
>
> We thank the reviewer for the thoughtful feedback and for appreciating the soundness of our theoretical framework. Below, we address the specific questions raised regarding the emergence of the hierarchy of details and the estimation of mutual information.
>
> **Emergence of Hierarchy of Details**
>
> We agree that understanding the hierarchy of emergent details is a crucial aspect of our work. The submitted paper already provides experimental indications of this phenomenon, such as the results obtained in Figure 3 on the Shapes3D dataset: the attributes that **occupy** the largest portion of the space (wall hue and floor hue) behave similarly and are the first to appear in the generation process, at $\tau=0.4$. The object hue emerges after these two, at $\tau=0.5$ but before orientation, scale and shape, which appear at $\tau=0.6$. Intuitively, given the nature of the dataset, this hierarchy is reasonable, since (for example) varying the orientation of a shape once the color is fixed, has a lower impact in terms of norms, than changing the color of an object (even if the orientation is changed, a reasonable portion of the pixels maintain approximately their value, whereas changing the color modifies all of them).
>
> To further corroborate our experimental findings, we are currently running experiments on the CelebA dataset, where we are observing a similar pattern of hierarchical feature emergence. For example, high-level attributes, such as gender, emerge earlier in the generative process than finer-grained details, such as the presence of eyeglasses, which only appear at later stages. We will place our additional results in the appendix of the camera ready version of the paper.
>
> **Estimation of Mutual Information**
>
> The estimation of mutual information (MI) in our framework is described in the main text and Appendix J. To aid the reviewer’s understanding, we summarize the key aspects of the process below:
> - Variance-Exploding Scheduler: A variance-exploding (VE) scheduler was selected for these experiments. This corresponds to the limit $\alpha \rightarrow 0$ for $m_t$​ (line 1100 in the appendix).
> - Denoisers and U-Net Architectures:
> The two terms in the square norm of Eq. (39) correspond to denoisers for the variable $V$, with or without extra information (the label).
> These denoisers are implemented using standard U-Net architectures, as detailed in Appendices J.2 and J.4.
> - Stochastic Process Dynamics: The stochastic process $Y_t$​ evolves according to generation dynamics similar to those described in Eq. (12). Using the unconditional denoisers, it is possible to simulate such dynamics by reparameterizing to obtain the score from the denoisers.
> - Numerical Integration: The integral in Eq. (39) is computed using the midpoint rule with 999 points for discretization (line 1235 in the appendix).
>
> This procedure ensures a robust and accurate estimation of mutual information, which plays a central role in validating our theoretical insights. We will consider including these details directly in the main text to make them more accessible to readers.

---

> > ### Comment · Reviewer_VD8o · 2024-12-02
> >
> > I acknowledge the authors rebuttal. I maintain my score.

---

### Official Review · Reviewer_PspT · 2024-11-03

**Soundness:** 2
**Presentation:** 1
**Contribution:** 2
**Rating:** 3
**Confidence:** 2

**Summary:**

The authors prove the mathematical equivalence between diffusion models and nonlinear filtering, where a stochastic process Y is controlled by an unobserved variable X. The authors show that diffusion generative modeling is related to inference of X using Y. If this is the case then the internal representations of denoiser networks must reflect our belief in X. They have empirical results demonstrating when certain latent statistics emerge during generation.

**Strengths:**

They rigorously define the connection between nonlinear filtering and diffusion.

**Weaknesses:**

The paper asks the reader to spend hours understanding their notations and results before describing the motivation in section 4. The paper is very challenging to read as is and I strongly recommend the authors clearly and succinctly describe their contributions before the technical details, which should also mostly be moved to the appendix.

While a connection between two mathematical frameworks is challenging to derive, it is only interesting if it leads to a deeper understanding of either technique. The fundamental claim of the paper -- that denoising networks must internally represent latent abstractions -- is seemingly obvious given that these networks are trying to predict the denoised state of Y. I would appreciate it if the authors could clearly articulate what they've learned about diffusion models in the language people currently use to describe diffusion models (if this paper is about diffusion models, by the way, perhaps the whole notation should be in terms of a diffusion model, not NLF).

**Questions:**

See weaknesses.

---

> ### Author Response · Authors · 2024-11-21
> **Rebuttal**
>
> We appreciate the reviewer’s constructive feedback. Below, we provide detailed clarifications and revisions that address the concerns raised.
>
> **Not Obviousness and Novelty of Our Contribution**
>
> To the best of our knowledge, this is the first work to investigate the emergence of latent abstractions in generative models from an information-theoretic perspective. While empirical studies have highlighted the progressive emergence of high- and low-level abstractions during the generation process, the theoretical underpinnings of this phenomenon have remained largely unexplored. Our framework fills this gap by extending the framework of nonlinear filtering (NLF) and provides a rigorous mathematical description of how such abstractions emerge and influence the generation process.
>
> This requires involved mathematical constructs, which we acknowledge can be challenging and require time to be appreciated, but that are essential to justify the gradual emergence of the posterior and its informational dynamics as a stochastic process. Importantly, our approach is general and does not only cater score-based diffusion models, which we use in our work as a simple illustrative example.
>
> We agree with the reviewer that score-based diffusion models provide an intuitive lens for understanding generative processes. However, our framework deliberately transcends this scope. Section 4 emphasizes that our findings apply to any generative model satisfying Assumption 1, including neural stochastic differential equations (SDEs) [1], Schrödinger Bridges [2], and stochastic normalizing flows [3]. These models exhibit dynamics where interpreting neural drift as denoising is not straightforward or may not even be feasible. For instance, in Schrödinger Bridges, the connection to a denoising score function is highly implicit. Similarly, in stochastic normalizing flows, the generative pathways often lack a clear denoising interpretation.
>
> While the concepts of denoising and score functions are widely accepted in the context of diffusion models, the precise mechanisms through which representations emerge at different stages of the generative process have not been previously formalized. For example:
> - It is well understood that score functions denoise data. However, it is not clear at which generation stage various details emerge or why simple (linear) classifiers can probe representations within neural network activations.
> - Our framework elucidates this by linking the emergence of latent factors to information-theoretic measures such as mutual information.
>
> **Emergence of Abstractions and Agreement Among Metrics**
> Our framework provides new insights into the hierarchical emergence of abstractions. Mutual information quantifies the influence of latent factors on observed data and reveals that global abstractions (e.g., semantics) emerge before localized details. This aligns with empirical findings in diffusion models and generalizes to other generative frameworks.
>
> Moreover, our work establishes why specific latent representations are encoded in the activations of generative networks. For example:
>
> - The sufficient statistics derived in our theoretical framework demonstrate that the posterior (a-posteriori measure) captures all relevant information about latent abstractions.
> - This explains why latent abstractions emerge gradually, supporting their hierarchical interpretation.
>
> A particularly compelling aspect of our empirical results is the exceptional agreement between:
>
> - **Mutual Information [our original contribution]**: Theoretical predictions of latent factor emergence based on information-theoretic measures.
> - **Entropy of Forking [our original contribution]**: Entropy reduction across forked generative pathways, which directly tracks the commitment of the generative process to specific latent abstractions.
> - **Linear Probe Accuracy**: High probing accuracy on features extracted from the denoising score network at corresponding noise levels.
>
> This agreement across three distinct metrics provides a robust empirical validation for our theoretical framework and supports its applicability to real-world generative models. Such consistency underscores the usefulness of our approach for analyzing the interplay between latent abstractions and observable data in generative models.
>
> We truly hope that the points above will help the reviewer to appreciate the extent of our theoretical contribution, and that the struggle and time spent to understand our paper (for which we deeply thank the reviewer) will not weigh negatively on the final assessment.
>
> [1] Kidger et al, Neural SDEs Made Easy: SDEs are Infinite-Dimensional GANs, ICLR 2021
>
> [2] De Bortoli et al, Diffusion Schrödinger Bridge with Applications to Score-Based Generative Modeling, Neurips 2021
>
> [3] Hodgkinson et al, Stochastic continuous normalizing flows: training SDEs as ODEs, UAI 2021

---

> ### Comment · Reviewer_PspT · 2024-11-23
> **Comment**
>
> Thank you for your response! I'd appreciate your elaborating further on the significance of this work. You state
>
>  >To the best of our knowledge, this is the first work to investigate the emergence of latent abstractions in generative models from an information-theoretic perspective.
>
> Formally describing the emergence of latent features is perhaps interesting to the information theory community, who may take for granted the importance of seeing if their theory can explain the behavior of diffusion models. But the larger ML community isn't necessarily surprised by the emergence of latent features; one may feel there is nothing to explain as the emergence of latent features is a fundamental behavior of almost all deep learning models (explained using "compression", PAC-Bayes, etc...). What would be nice is if you could advocate for the utility of the information theoretic perspective: does this theory have the potential to help us build better models? Does it offer a satisfying explanation of previously surprising empirical results? Does it suggest new experiments? Does it give us predictive power over the behavior of these models that we didn't have before?

---

### Author Response · Authors · 2024-11-25

In response to some comments concerning the applicability of our framework to some more realistic datasets, we have uploaded a new version of the paper where we include preliminary results on the CelebA dataset (Appendix J.7).

We thank again the reviewers and the AC for their time and their feedback.

Regards,

The authors

---

### Meta-Review · Area_Chair_3fMf · 2024-12-19

**Metareview:**

The paper 'Latent Abstractions in Generative Diffusion Models' was reviewed by 4 reviewers who gave it an average score of 5.25 (3+6+6+6). The reviewers gave merit to the paper for the clear motivation, rigour in presentation, and interesting viewpoints. On the other hand, the reviewers were less enthusiastic about the rather technical presentation, lack of connection to practical benchmark problems/demonstrations, and disconnection from a presentation that would benefit the wider ICLR audience. Even if the majority of the reviewers end up giving this submission a 'borderline accept' score, the reviewer consensus is that the paper is not quite fit for ICLR in its current form.

**Additional Comments On Reviewer Discussion:**

The reviewers interacted with the authors during the discussion period. However, the scores did not change during the discussion period. The authors updated their work based on reviewer feedback, but I'm afraid these updates would have been needed already in the original submission phase.

---

### Decision · Program_Chairs · 2025-01-22

Reject